# Divergent Deborah number-dependent transition from homogeneity to heterogeneity

Dan Xu [1], Yang Yang [1], Lukas Emmerich [2], Yong Wang [3] & Kai Zhang [1] ✉

Heterogeneous structures are ubiquitous in natural organisms. Native heterogeneous structures inspire many artificial structures that are playing important roles in modern society, while it is challenging to identify the relevant factors in forming these structures due to the complexity of living systems. Here, hybrid hydrogels consisting of flexible polymer networks with embedded stiff cellulose nanocrystals (CNCs) are considered an open system to simulate the generalized formation of heterogeneous core-sheath structures. As the result of the modified air drying process of hybrid hydrogels, the formation of heterogeneous core-sheath structure is found to be correlated to the relative evaporation speed. Specifically, the formation of such heterogeneity in xerogel fibers is found to be correlated with the divergence of Deborah number ($De$). During the transition of $De$ from large to small values with accompanying morphologies, the turning point is around $De = 1$. The mechanism can be considered a relative humidity-dependent glass transition behavior. These unique heterogeneous structures play a key role in tuning water permeation and water sorption capacity. Insights into these aspects can prospectively contribute to a better understanding of the native heterogeneous structures for bionics design.

Living things are famous for their adaptive multifunctions in various environments, which can be attributed to the basic principle of structure-determined functions[1-3]. Different from homogeneous structures universally found in artificial items, organisms generally possess complex, heterogeneous structures so as to acclimatize themselves to complicated and changing environments. For instance, the heterogeneous structure in mollusk shells called 'brick-and-mortar' endows the invertebrates with superior toughness and strength using limited resources, providing an inspiration to construct high-performance materials[4,5]. In contrast, such heterogeneous structures with advanced performance are not easy to be realized in most artificial materials. The heterogeneous structures spontaneously formed in plants are quite popular as well, such as the organization of trunks[6] and leaves[7], in which the evaporation process plays a crucial role in the transition of these structures[8]. Thus, it is essential to understand the formation of heterogeneous structures in nature, which will not only provide new aspects about the structure transition in living things but also accelerate the development of material science.

Facing the challenges of complex systems in living things, it is quite difficult to figure out the origination of heterogeneity. Hydrogels, as a typical material with high water content and crosslinked networks, are considered to have analogous structures and properties as organisms[9,10]. Benefiting from the application-oriented research works, various hydrogels aiming to promote health care covering cell scaffold[11], artificial muscle[12], cartilage[13], vascular networks[14] and wound compress[15], in addition to many other applications, have been

[1]Sustainable Materials and Chemistry, Department of Wood Technology and Wood-based Composites, University of Göttingen, Büsgenweg 4, D-37077 Göttingen, Germany. [2]Department of Wood Biology and Wood Products, University of Göttingen, Büsgenweg 4, D-37077 Göttingen, Germany. [3]Laboratory for Fluid Physics, Pattern Formation and Biocomplexity, Max Planck Institute for Dynamics and Self-Organization, Am Faßberg 17, D-37077 Göttingen, Germany. ✉e-mail: kai.zhang@uni-goettingen.de

developed. Considering the alterable and adjustable physicochemical process during the dehydration of hydrogels, the advantageous properties of hydrogels, especially as networks with tunable flexibility and viscosity as well as tolerance of versatile components, allow them to be highly suitable and comparable candidates for revealing the general principle for the formation of heterogeneous structures. However, using hydrogels as an open system to investigate the transition of heterogeneous structures that could inspire the future construction of materials with unique functions is not known.

On the other hand, although nanocellulose has been widely used to improve the performance of composite materials due to its superior mechanical properties[16] or to impart stimuli-responsive behaviors via its wetting properties[17,18], or to develop the approaches[19,20] and strategies[21] to realize the alignment of nanocellulose in bulk solutions or composite materials, the main focus of the current work lies on the structure formation and development of the materials depending on their intrinsic rheological properties. Moreover, this study sheds light on understanding the effect of such a dynamic process, i.e., the dehydration of composites in the environment, on the resulting structures of materials. In this work, a hybrid hydrogel containing embedded CNCs was used to investigate the formation of heterogeneous structures in the air drying process, with plants as heterogeneous objects, thanks to the superior deformation ability of hydrogels and the feasible organization of CNCs within hydrogels[22,23]. In particular, the anisotropic optical property of CNCs will help us to trace the transition of microstructures[24]. After uniaxial stretching of hydrogels, the gradual formation of spontaneously shaped heterogeneous structures during the following air drying and finally in the dried xerogel fibers was continuously observed using polarized optical microscopy (POM) and scanning electronic microscopy (SEM). The relative evaporation speed of hydrogels was also tuned to simulate the various dehydration conditions.

## Results

Inspired by the growth and the heterogeneous structures of plants (Fig. 1a), hybrid polyacrylamide hydrogels with embedded stiff CNCs were prepared using dynamically crosslinked flexible polymer networks as matrix (Fig. 1b, Supplementary Fig. 2). Based on TEM images of 300 individual CNCs after TEMPO-mediated oxidation, CNCs of $209 \pm 55$ nm in length and $14 \pm 5$ nm in width were prepared, with an aspect ratio of ~15 (Supplementary Fig. 3). The content of carboxyl groups in CNCs was determined to be approximately 1.06 mmol/g by conductivity titration, endowing CNCs with a zeta potential of $-46.40 \pm 1.16$ mV (Supplementary Fig. 4). Resulting stable and well-dispersed CNC suspensions could be integrated into the polymer networks of hybrid hydrogels after the photo-initiated polymerization. The typical viscoelastic behavior was reflected by the rheological properties, which also demonstrate the flexibility and dynamics of the hybrid hydrogels (Supplementary Fig. 5). Homogeneous porous structure of freeze-dried hydrogels, as shown in the SEM image, indicates the homogeneity of as-prepared hydrogels and thus uniform distribution of CNCs (Supplementary Fig. 6). In addition to improved mechanical properties associated with CNCs, the instinct anisotropic structure of CNCs with the accompanying birefringent properties can be utilized to uncover the structural features in hybrid materials[23].

The procedure for preparing xerogel fibers from hybrid hydrogels is illustrated in Fig. 1b, which consists of uniaxial stretching and a subsequent air-drying process. It is worth emphasizing that the evaporation process in such cylindrical samples is the isotropic process along the radial direction, thus avoiding the intrinsic heterogeneity induced by the radial phenomenon (more details in Supplementary Fig. 7). These two steps can be seen as a generalized process that simulates the growth of a tree in its natural environment. The slightly brown color of obtained xerogel fibers was caused by the oxidation of catechol moieties (Fig. 1c). Interestingly, we found a prominent heterogeneous structure in the dried xerogel fibers, which is different than the homogeneous porous

structures in virgin hydrogels (Fig. 1d and Supplementary Fig. 5). Specifically, the inner part was porous, while the outer part of the xerogel fibers was denser. A clear boundary is present and separates these two parts, which resembles the bark and pith in tree trunks.

We further analyzed the structural properties of xerogel fibers regarding the microstructures and optical properties. CNCs within the polymer matrix are ordered during both the uniaxial stretching and subsequent air drying process[23], while the alignment of polymer chains was eliminated due to the fast cleavage and rebinding of dynamic bonds between borate and catechol moieties (Supplementary Fig. 2)[25,26]. Then, the xerogel fibers were observed under two main angles, one as an oblique section (Fig. 1e) and the other as a vertical cross section (Fig. 1h). While corresponding SEM images of these cross sections showed similar porous microstructures, the color variations viewed between crossed polarizers indicate different organized structures based on the alignment of CNCs (Supplementary Figs. 8 and 9).

The entire xerogel fibers showed evenly cyan color under orthogonally polarized light due to the alignment of CNCs along the stretching direction (Fig. 1e), indicating the uniform arrangement and distribution of CNCs in the interior. In comparison, at least three continuous orders of colors are notable for the oblique cross section under the anamorphism of thickness (Fig. 1f, g), according to Michel-Lévy interference chart[27]. The presence of these continuous orders of colors indicates the nearly linear correlation of the retardation to the thickness, while the birefringence of the xerogel fibers remains similar based on their uniform structure (retardation = thickness × birefringence). It should be noticed that the interface between different color zones was meniscus, and this phenomenon is mainly induced by distinct thicknesses on curved surfaces.

In comparison, the vertical cutting section interestingly showed a heterogeneous structure with separated inner and outer parts, as observed on the slices of xerogel fibers with a thickness of ~100 μm (Fig. 1h). In the inner part, a lot of bright spots are visible, which should be induced by the self-assembled CNCs in short-range, e.g., self-assembled tactoids and clusters[28,29]. Around this interior part, an outer part existed as a dark ring without obvious light spots, so these self-assembled structures of CNCs should be absent in the outer layer. To eliminate the effect of curved surface on cylindrical samples, a xerogel film obtained by using cuboid hydrogel with the same procedure was also observed (Supplementary Fig. 10). Due to the lower geometric symmetry from cuboid to cylinder, the colors of xerogel film under polarized light changed from violet (central zone) to sky blue (edge zone), also indicating the different modality of CNC organization in different zones. The SEM images of such xerogels with rectangular cross section revealed a heterogeneous structure as well (Supplementary Fig. 11).

By inserting a full waveplate retarder to enhance the phase contrast of POM images, the inner region was coral, while the outer layer showed a dark blue color (Fig. 1i, j). Hence, xerogel fibers contain, after the uniaxial tensile and air drying process, a heterogeneous structure with two diverse areas as constituting modalities along the longitudinal direction of xerogel fibers: the inner part as the core and the outer part as the sheath.

In order to dispel the heterogeneity in the air drying process resulting from the geometry parameters, three types of hydrogels of the same size while containing the mediums with different evaporation speeds were used for the fabrication of xerogel fibers. In detail, the buffer solution (buffer), a faster evaporation solution as a mixture of buffer/10% ethanol, and a slower evaporation solution as a buffer containing 5% glucose were applied (Supplementary Figs. 1 and 15). Although all three samples led to porous structures as the major areas, the flat and smooth outer layer appeared only in xerogels from buffer and the mixture of buffer/10% ethanol (Supplementary Fig. 12). For these two samples, the thickness of the sheath layer reached up to 6.2 and 11.4 μm, respectively (Fig. 2a). In contrast, the sheath layer did not exist in xerogel fibers from buffer with 5% glucose with a lower

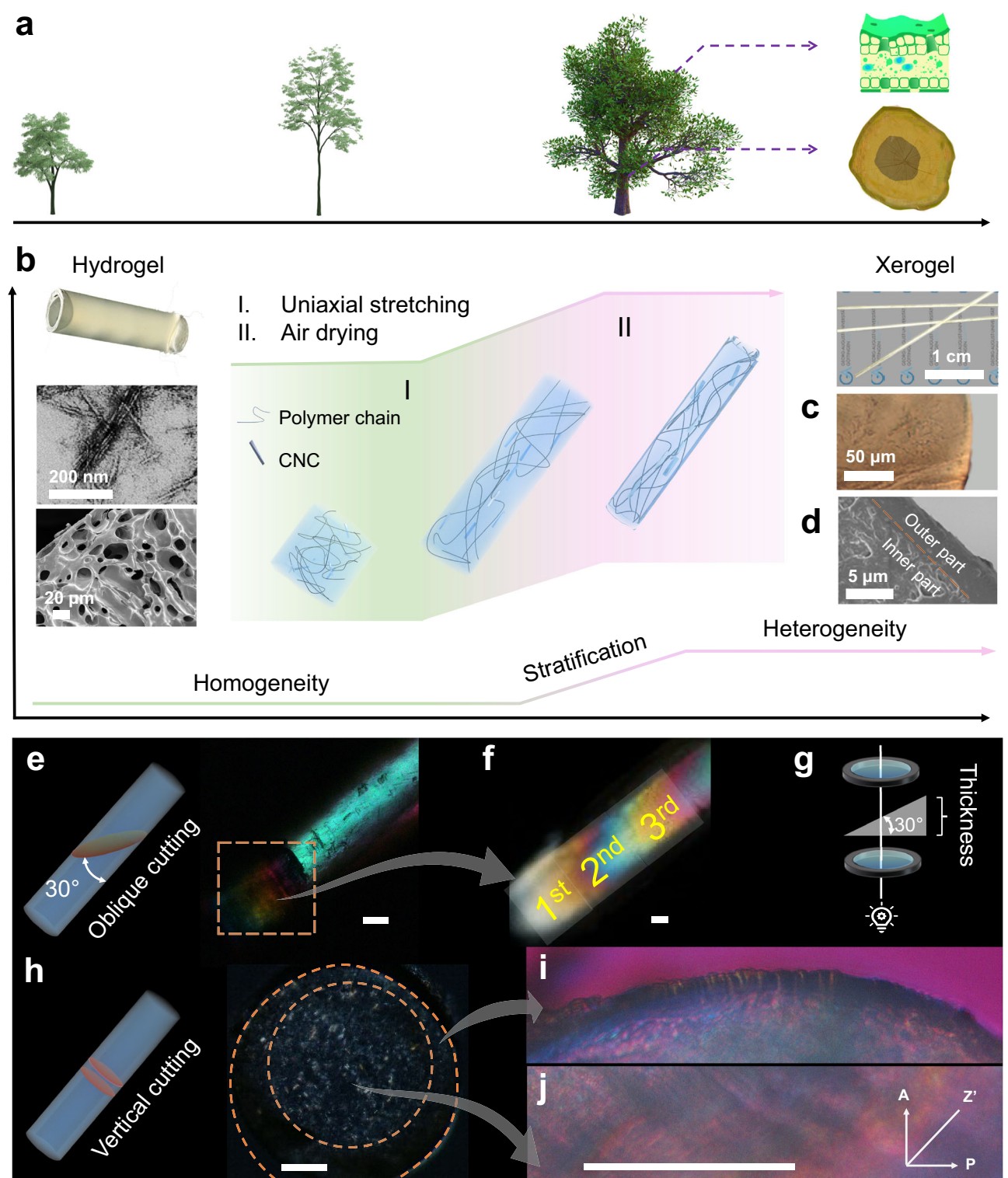

**Fig. 1 | Biomimetic heterogeneity generated in the transition process from hydrogel to xerogel. a** Representative heterogeneous structures found in the leaves and trunks of trees in nature. **b** Illustrated process to prepare xerogels from CNC hybrid hydrogels; the virgin hydrogel was of homogeneous, porous structure. **c** Optical microscope and **d** SEM image of the cross section of xerogel fibers. **e** Overall POM images of a xerogel fiber. The inset is the schematic illustration of cutting a bevel section of a xerogel fiber. **f** POM image of the bevel section. **g** Illustration for showing the light transmitting the bevel section. **h** POM images of the vertical cross section. The inset is the schematic illustration of forming the vertical cross section of a xerogel fiber. **i** POM images (with 1/4 λ waveplate) of the enlarged inner part and **j** outer part. The scale bars in POM images are 100 μm.

evaporation speed and a uniform rough cross section was found. This result indicates that, as with cylindrical hydrogels, the heterogeneity in the resulting xerogels is dominated by the distinct air drying process rather than the geometry-induced anisotropic air drying.

The thickness of outer layers in these different xerogel fibers and their residual water contents were then analyzed (Fig. 2b). The values of T/R were used to characterize the morphology evolution of these two different regions within obtained fibers (T: thickness of outer part;

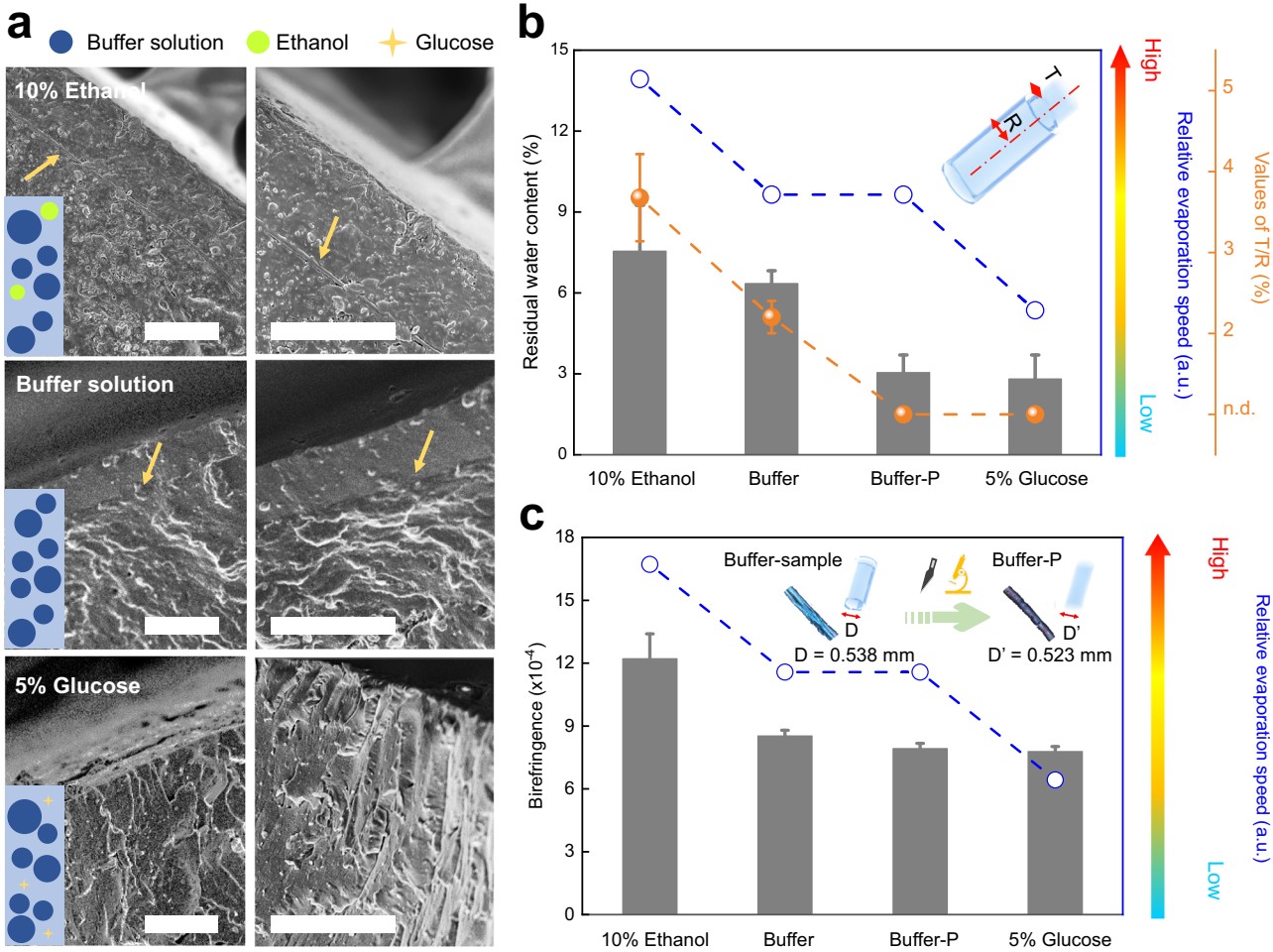

**Fig. 2 | Transition of xerogel fibers from homogeneity to heterogeneity in various mediums. a** SEM images of the vertical cross section of xerogel fibers from various media to simulate the relative evaporation difference in nature. The colored arrows indicate the boundaries of the regions with distinct structural features. Scale bars: 10 µm. **b** Residual water contents and heterogeneous structures in xerogel fibers depending on the evaporation speed (n.d.: the clear sheath structure was not found using SEM; $n = 3$ independent experiments). Buffer-P means the sheath layer of samples was peeled off. Inset: schematic illustration for the geometrical parameters of heterogeneous fibers. **c** Birefringence values of xerogel fibers from various solutions with different evaporation speeds ($n = 3$ independent experiments). Inset: POM images of the xerogel fiber from the buffer and peeled xerogel fiber (buffer-P). Error bars are SD.

R: radius of xerogel fiber. More details in Supplementary Fig. 13). While no outer layer is present in 5% glucose samples, the T/R values of 10% ethanol and buffer fibers were calculated to be $3.7 \pm 0.5\%$ and $2.2 \pm 0.2\%$, respectively. By correlating T/R values and residual water content, 10% ethanol, buffer and 5% glucose fibers with decreasing T/R values showed a similar trend with corresponding decrescent residual water content of 7.5%, 6.4% and 2.8%, respectively (Fig. 2b). The outer layer of xerogel fibers can even be peeled off from the inner region, e.g., using xerogel fibers from buffer (referred to as buffer-P). The remaining inner region shows a similar, entirely homogeneous structure as resulting xerogel fibers from buffer containing 5% glucose. Interestingly, these fibers without dense outer layer had almost the same residual water contents, regardless of their formation surroundings with distinct relative evaporation speeds. Without the outer layer, buffer-P fibers, after peeling off the outer layer from buffer fibers, only contained a residual water content of 53.1% after equilibration. These results indicated the important role of the outer layer in preventing water exchange, which can be critically important in practical applications.

Furthermore, the arrangement of anisotropic CNCs within xerogel fibers was further analyzed to elucidate their structure difference by determining their birefringence, which represents the orientation information[30,31]. Since the alignment of polymer chains can be eliminated by the reorganization, the birefringence of resulting materials was dominated by the orientation of CNCs[23]. The birefringence values of xerogel fibers from 10% ethanol, buffer, and 5% glucose solutions were measured to be $0.00122 \pm 0.00012$, $0.00085 \pm 0.00003$, and $0.00078 \pm 0.00002$, respectively (Fig. 2c). Therefore, a better organization of CNCs is even present within the thicker outer layers. The self-assembled clusters of CNCs in the inner part could be accountable to the lower orientation index. By removing the outer layer, the birefringence value of buffer-P samples was measured to be $0.00079 \pm 0.00003$, which dramatically decreased by 7% compared with buffer samples. This result further verifies the relatively higher orientation index for CNCs in the outer part and the lower orientation index for the inner part. Moreover, this highly similar birefringence of buffer-P fibers and 5% glucose fibers (without an obvious outer layer) also demonstrated a similar organization modality of CNCs and therefore a similar structure formation process.

After the initial uniaxial stretching leading to partial alignment of the CNCs (Fig. 3a), a faster evaporation process with the relatively higher evaporation speed as for xerogel fibers from buffer or buffer/ 10% ethanol could have induced a denser outer layer, resulting in an apparent heterogeneous sheath-core structure. At the same time, the rapid evaporation of the solvent from the outer layer promoted the

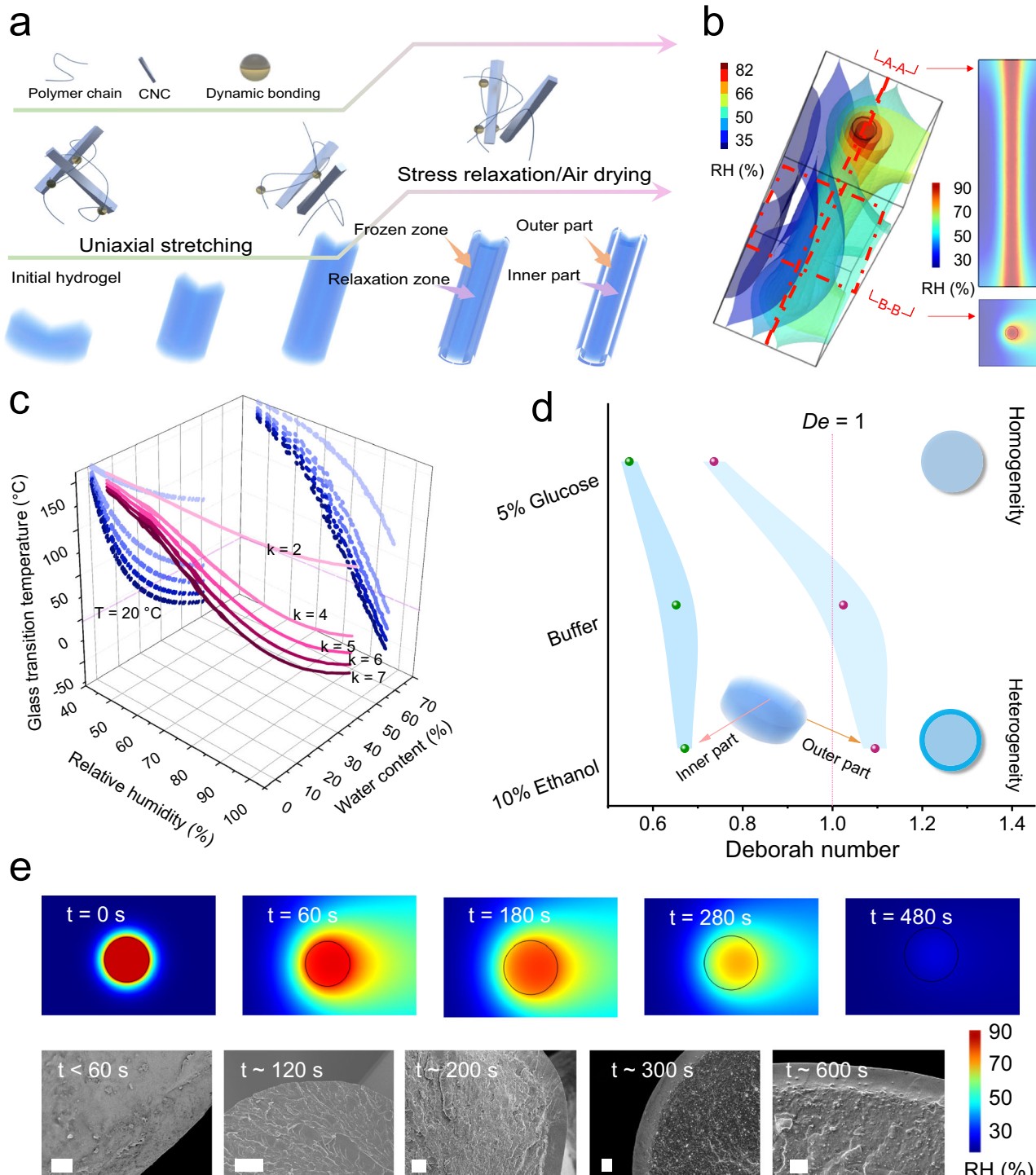

**Fig. 3 | Hydrogel-xerogel transition process during air drying. a** Schematic illustration of the typical stress relaxation process of hybrid hydrogels and the stratification development. **b** Snapshot of RH distribution around hydrogels in an intermediate air drying stage (after 200 s). **c** Glass transition temperatures of hydrogel fibers in surroundings with various RH or with various water contents according to the Gordon-Taylor equation, the values of RH or water contents are recorded from simulation results. **d** Deborah number of various samples according to the time point when the water content was less than 20% for their inner and outer parts ($n = 3$ independent experiments to determine relaxation time). **e** RH distribution around hydrogel fibers at various times and corresponding SEM images, and the Deborah number was calculated according to the time points labeled on SEM images. Scale bars: 10 μm. Error bars are SD.

alignment of the CNCs in the outer layers so that a higher orientation index of the CNCs in the outer part was generated than that in the inner part. In comparison, a slower evaporation process, as by using a buffer with 5% glucose or with the formation of the outer layers that hamper the evaporation, leads to more porous structures with lower orientation of CNCs.

Since the alignment of CNCs was different in the outer and inner regions, we further analyzed the evolution of polymer networks inside the hybrid hydrogels along with the air-drying process. As shown in Fig. 3a, the uniaxial stretching with external force induced the partial orientation of polymer chains and CNCs, while the reorganization of the dynamic polymer network as a stress relaxation stage counted

against the ordering of CNCs. During air drying, different relaxation in the outer and inner layers will cause their different properties. The stress relaxation of soft materials containing water was deeply influenced by water content and related relative humidity (RH)[32–34]. Herein, the air drying process of hydrogel fibers, as the key operation in the formation of xerogels, was investigated numerically with the finite element method at first (Supplementary Table 1)[35,36]. The simplified physical models were used (Supplementary Fig. 14a). Then, the gradient RH and water content distribution were verified from the external environment to the interior of hydrogels in the air-drying process (Fig. 3b, Supplementary Movie 1). The flow of air dehydrated the hydrogel surface layer (drying front), while the inner part could maintain the high RH for a long time and thus undergoes a slow drying process, although the whole hydrogel fiber reached the same RH after enough time (usually more than 300 s).

$$T_g = \frac{kw_1 T_{g1} + w_2 T_{g2}}{kw_1 + w_2} \tag{1}$$

In view of the mobility of polymer chains in forming xerogels, the glass transition temperature was considered. Based on free volume theory (schematic illustration in Supplementary Fig. 16), the water sorption induced plasticization of xerogel fibers was quantitatively analyzed using the Gordon-Taylor equation[37,38], in which the RH-water content-glass transition temperature was correlated. Equation 1 shows the calculation, where $w_1$ and $w_2$ are the weight fraction of water and polymer, $T_{g1}$ and $T_{g2}$ are the glass transition temperature of water (138 K) and polymer, respectively[39]. The glass transition temperature for polyacrylamide is set at around 436 K[40]. Factor $k$ is an empirical parameter indicating the strength of water plasticization, which usually ranges from 1 to 10 for hydrophilic polymers[37,41,42]. During the air drying stage, water contents in hydrogels decreased until the RH inside the fibers reached environmental RH. On the other hand, the xerogel fibers exchanged water with the external environment, and they could absorb/desorb water due to variant RH. Thus, RH and water content can be considered as the correlation moments, and the values were obtained according to simulation results. It should be noted that the values matched well with experimental results, e.g., the water content was around 20% when RH = 80% according to dynamic vapor sorption (DVS) measurements (Supplementary Fig. 35).

The glass transition temperature dramatically decreased with rising water content, indicating higher mobility of polymer chains (Fig. 3c). Because the storage modulus of obtained xerogel fibers under different RH markedly decreased ($T$ = 25 °C), when RH was up to ~80% (Supplementary Fig. 17), $k$ = 6 would be the reasonable description of hydroplastic behaviors of our system. Herein, the calculated $T_g$ of the xerogel fibers was around 20 °C at the RH of ~80%, while it could exceed 95 °C at the RH near 30% (close to normal environmental RH). This also explains the phenomenon that the curved xerogel fibers were found under conditions with high environmental RH (Supplementary Fig. 17).

Deborah number (De) was then introduced to investigate the generation of heterogeneity in xerogels, in which both the relaxation time and interaction time were identified:

$$De = \frac{\tau}{t_{int}} \tag{2}$$

where $\tau$ is the relaxation time and $t_{int}$ is the interaction time[43]. The Voigt-Kelvin model was used to describe the stress relaxation behaviors of polymer networks, and the $\tau$ was obtained from fitted results (Supplementary Fig. 18)[44,45]. In combination with the distinct distributions of RH and water contents during the formation of xerogel fibers, we comprehensively compared the stress relaxation behaviors in the inner part and outer part (Fig. 3d). Regardless of the mediums, the hydrogels behaved with similar mechanical properties (Supplementary Fig. 19). As we mentioned before, the mobility of polymer chains remains sufficiently high at around room temperature when the water content was over 20%. Taking the time between the beginning of drying and the point where water content reached 20% ($t_{0.2}$) as the interaction time, the disparate De with significance was obtained. The same dynamic crosslinked network endowed different samples with similar relaxation time (~210 s), while the various mediums inside varied the evaporation speed of samples (Supplementary Fig. 20). Specifically, for 10% ethanol samples with a higher evaporation speed, the water contents were lower than 20% for the outer part and inner part at $t_{int}$ of 190 and 310 s, respectively. As a result, De < 1 happened in the inner part while for outer part was obvious of De > 1. Meanwhile, the heterogeneous structures were found in formed xerogel fibers. In contrast, for the 5% glucose sample De < 1 were present for both the inner and outer part, resulting in the homogeneous xerogel fibers. For the xerogels with heterogeneous core-sheath structures originating from buffer samples, De was about 1.02 for the outer part, and De was about 0.65 for the inner part, indicating that the divergence of De occurred together with the emergence of heterogeneity in the xerogels as well. Therefore, such a transition from homogeneous to heterogeneous structure and the formation of heterogeneity is highly correlated with the divergent De values adapted to the hydrogels.

It is considered that De is more than 1, the material is relatively solid; if De is lower than 1, the material is relatively liquid[43]. In this system, the mechanism for this divergent De-determined transition from homogeneity to heterogeneity can be proposed as follows. As for the inner part or at certain conditions with De < 1, the interaction time was longer than the relaxation time, so the polymer chains at transit equilibrium and have sufficient time to reorganize, leading to less ordered structures (Fig. 3a). For the outer part with De > 1, polymer chains were quickly frozen due to fast dehydration, so that the relatively well-ordered CNCs could be reserved, thus the orientation degree in the outer part was higher than that in the inner part. For example, the xerogels from 10% ethanol and buffer samples contain such heterogeneous core-sheath structures. In comparison, the xerogel fibers from 5% glucose samples were of homogenous structure and lower orientation degree of CNCs. By knowing the effect of the De, the heterogeneous structures were also fabricated in non-prestretched xerogel fibers by tuning the interaction time (Supplementary Fig. 21). Specifically, these xerogel fibers can be prepared by undergoing the creep and air drying process so that the stratification was induced by distinct arrangement process of polymer networks during the whole preparation.

The dynamic process of generating heterogeneity in the process can be systematically verified. Here, we took the xerogel fibers originating from buffer solution with the $\tau$ = 213.0 ± 7.2 s as an example (Fig. 3e). The SEM images after the lyophilization of fibers revealed that the microstructure of the hydrogel fibers was homogeneous in the early stage of air drying, for instance at the evaporation time less than 60 s, which was consistent with the output of simulation. Stratification appeared and the sheath layer emerged after ~200 s evaporation. With the increasing evaporation time, the heterogeneity was also enhanced. The T/R values increased from almost 0 (not detectable) to 1.9% and 2.2% when evaporation time changed from 60 s to 300 s and 600 s, respectively. In contrast, as-prepared hybrid hydrogels (without stretching and air drying) only contained the typical porous structure with homogenous distribution, with the same thickness of boundary layer and internal pore walls (Supplementary Fig. 6). Therefore, by defining the interaction time, we demonstrated that the key parameter to tune the formation of heterogeneity in xerogel fibers was the transition of De from large to small values. Moreover, the turning point was around De = 1.

The transition from homogeneity to heterogeneity in the air-drying process of hydrogels was also studied under different

conditions on samples of various shapes (Supplementary Figs. 22–26). The homogeneity and heterogeneity in the resulting xerogels also follow the divergent *De*-induced structural transitions.

The heterogeneous structures originating from the organization modality of polymeric hydrogel networks were further demonstrated by humidity sweep tests on dynamic thermal mechanical analysis (DMTA). The xerogels without sheath layers, such as 5% glucose and buffer-P fibers, exhibited a single glass transition with increasing RH (Supplementary Fig. 27). The isothermal sorption curves of xerogels showed a typical type B hysteresis loop[46–49] (Supplementary Figs. 28, 34, 35a–c), indicating the presence of slit-shaped pore structure. This is also in agreement with the formation of xerogel fibers, in which CNCs were organized along the stretching direction while the porous structures were formed (Fig. 3a).

In order to better illustrate the water vapor sorption and desorption behaviors of heterogeneous structures, obtained xerogel fibers were systematically investigated under constant RH (Supplementary Fig. 29). Here, the xerogel fibers prepared from buffer were taken as examples. A typical water vapor sorption and desorption process under constant RH is shown in Fig. 4a. The water vapor could penetrate into xerogel fibers via both surface and cross section, and the geometric parameters of xerogel fibers were evaluated by aspect ratio (L/D). Xerogel fibers with 3 different geometries were used: L/D of 38.1, 9.4 and 3.5. They have similar diameters but different lengths showing distinct water-holding capacities (Supplementary Table 2,

Supplementary Figs. 30–32). Regardless of the geometrical parameters, the equilibrium water contents of these xerogel fibers were almost the same around ~21.4% (Fig. 4b). Therefore, the water holding capacity was mainly dominated by the microstructures of the xerogels themselves. In contrast, if the outer layer of xerogel fibers was peeled off, the equilibrium water content increased to 23%. This demonstrates the higher water-holding capacity of the inner part of the xerogel fibers than the outer layer. The higher degree of orientation by CNCs in the outer layer causing less accessible surface area could be partially the reason for the lower water capacity[50]. To accurately show the time to reach equilibrium status, we normalized the time by dividing the actual time by the mass of the corresponding sample. Obviously, the decrease in L/D would accelerate the water transmission process (Fig. 4c), and the xerogel fiber with the L/D = 3.5 exhibited the fastest sorption-desorption speed. Moreover, dislodging the outer layer of xerogel fibers apparently promoted the speed of water transmission. E.g., xerogel fibers with L/D = 38.1-P needed much less time to reach equilibrium compared to xerogel fibers with L/D = 38.1, indicating the barrier function of the outer layer toward water vapor transmission.

The water sorption process of xerogel fibers can be quantitatively described using three common models, pseudo-first-order model (Eq. 3), pseudo-second-order model (Eq. 4) as well as Weber and Morris intraparticle diffusion (IPD) model (Eq. 5)[51–53].

$$ln\left(Q_e - Q_t\right) = lnQ_e - k_1 t \qquad (3)$$

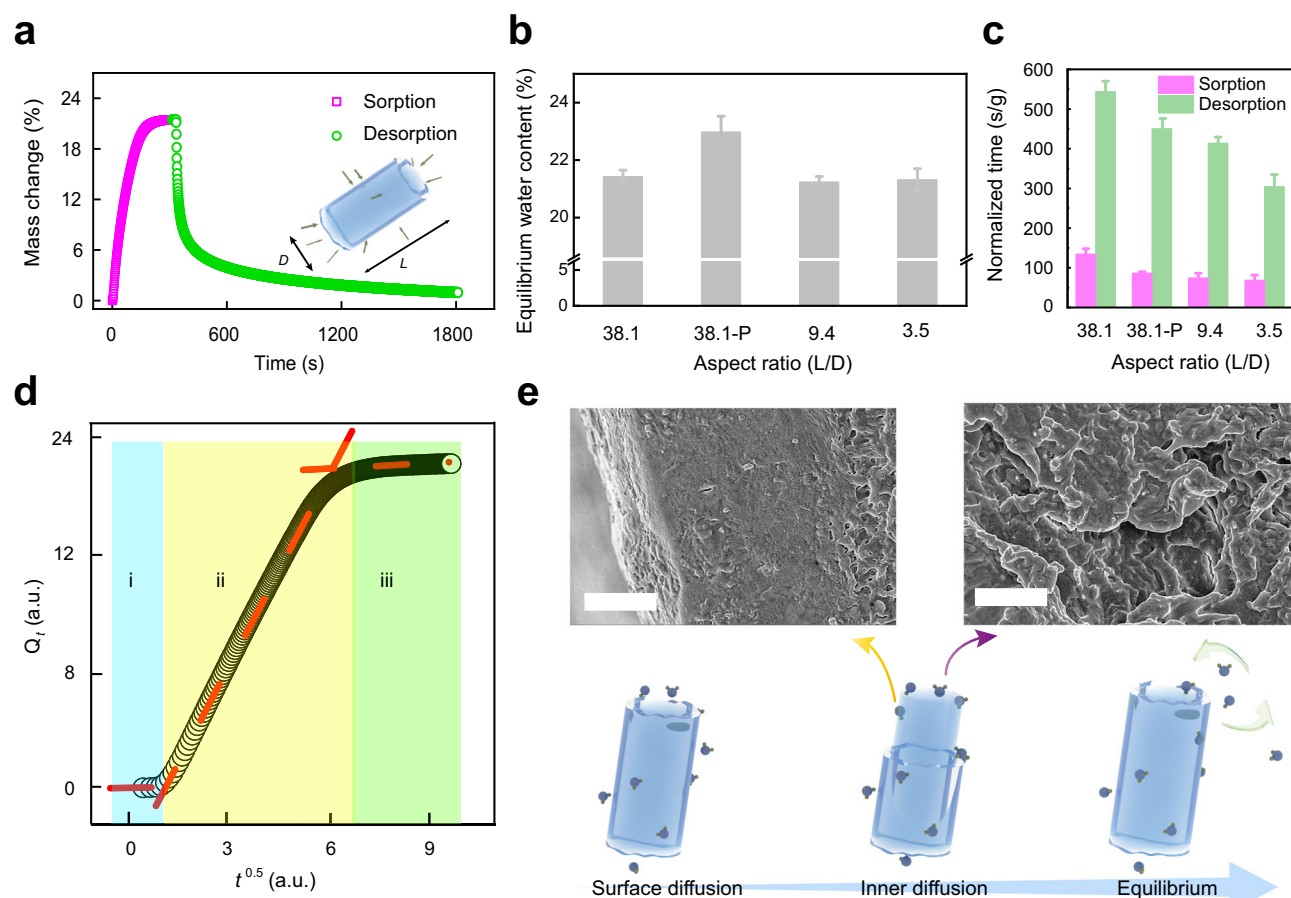

**Fig. 4 | Water vapor sorption/desorption behaviors of various xerogel fibers.** **a** A typical sorption-desorption curve of xerogel fibers. Inset: schematic illustration of water vapor diffusion into xerogel fibers. **b** Equilibrium water content of xerogel fibers (*n* = 3 independent experiments). **c** Normalized time for various samples to reach the equilibrium status through the water vapor sorption-desorption process (*n* = 3 independent experiments). **d** The kinetic analysis of the water vapor sorption process via intraparticle diffusion model. **e** SEM images of different areas of a xerogel fiber at distinct stages of water vapor sorption. Scale bars: 2 μm. Note that 38.1-P means the sheath layer of samples was peeled off. Error bars are SD.

$$\frac{t}{Q_e} = \frac{1}{k_2 Q_e^2} + \frac{t}{Q_e} \tag{4}$$

$$Q_t = k_{id}\sqrt{t} + C \tag{5}$$

where $t$ means the sorption time, $Q_e$ means the adsorption capacity at equilibrium, and $Q_t$ means the adsorption capacity at time point $t$.

It was found that both the pseudo-first-order model and pseudo-second-order model fitted well with experimental results (Supplementary Fig. 33, Supplementary Table 3). First, the $k_1$ parameter, as the time-scaling factor the value of which decided how fast the equilibrium in the system can be reached, was demonstrated to increase from 0.0627 to 0.1178 due to the decrease of L/D, in agreement with the observational results. Then, the $k_2$ constant value of the xerogel fibers, which was usually strongly dependent on the applied initial solute concentration, was almost the same (~0.002) due to the same applied RH. The IPD model also well described the water sorption process of xerogel fibers when RH = 80% (Fig. 4d). The intraparticle diffusion constants ($k_{id1}$, $k_{id2}$ and $k_{id3}$) were related to different stages of water vapor sorption. The change of $k_{id1}$ in stage $i$ was induced by the surface sorption of water vapor. The $k_{id2}$ in stage $ii$ was higher than $k_{id1}$ in stage $i$, which could be attributed to the internal slit-shaped pores. Meanwhile, the increasing $k_{id2}$ with lower L/D values indicates the hampering effect of the existing outer layer on the water transmission (Supplementary Table 3), which is consistent with the above-mentioned result. Stage $iii$ was also found when the sorption of the porous structure approached saturation. In comparison with $k_{id2}$, the values of $k_{id3}$ were much lower. Based on these results, the water vapor should have been adsorbed on the surface of xerogel fibers at first (Fig. 4e), while the surface diffusion could largely affect the total speed of water transmission. Then, the inner diffusion determined the overall time for xerogel fibers to be saturated until the equilibrium. At the same time, the heterogeneous structures with the existing outer layer promoted the water-holding capacity of xerogel fibers, which will prevent fast loss of water with great practical potentials (Supplementary Fig. 35). Regarding the hysteresis loop, the mathematic area increased with the increasing sheath layer, demonstrating the important role of sheath layer in tuning water sorption-desorption process as well. Furthermore, Pore Size Distribution (PSD) of the xerogel samples was analyzed via the BJH method. It was found that the xerogels (10% ethanol and buffer) with heterogeneous structures have primary and secondary pore diameters, which are centered at 2.6 nm and 8.1 nm. This result indicates the distinct pore structures for outer part and inner part (Supplementary Fig. 36). In contrast, the homogenous xerogels (5% glucose) show that the primary, secondary and tertiary pore diameters are centered at 2.6 nm, 4.5 nm and 8.1 nm, respectively. Moreover, the distributed frequency ($d$Vp/$d$Rp) of the smallest pores is quite low, which may explain the undetected outer layer in the 5% glucose xerogel fibers. The same largest pore diameter for all these samples indicates a similar porous structure inside. The total pore volumes of different samples also indicate that the disappearance of the outer layer results in an increase in pore volume. Therefore, the higher density of the outer layer (1.23 g/cm³) and the lower inner layer (1.11 g/cm³) should be induced by the different pore structures and total pore volumes (Supplementary Fig. 37).

## Discussion
In summary, inspired by the formation of heterogeneous structures of natural plants, xerogel fibers with heterogeneous structures were prepared after sequential uniaxial stretching and subsequent air drying. The observation reveals distinct properties of the inner part as core and outer layer as a sheath in one single sample. The outer layer was dense with stronger orientated microstructures, while the inner part was porous and uniform. By adapting $De$ to the air drying of hydrogels as interaction time, the formation of heterogeneous structures can be correlated with the divergent $De$, while tuning $De$ values can trigger the transition from homogeneity to heterogeneity in xerogel fibers. Specifically, the turning point was around $De = 1$. Combining numerical modeling and experimental investigation, the mechanism behind the association between $De$ and heterogeneity is water content-dependent glass transition. By quantitatively analyzing the results in DVS tests, this unique heterogeneous structure was demonstrated to be the key parameter in tuning the water transmission and the capacity of water holding. Herein, based on the physicochemical process in the modified dehydration process of dynamic hybrid hydrogels, we provide a new angle to understand the formation of heterogeneous structures in soft matter. A number of fields, including bionics design, soft-bodied robots and even tissue engineering, will draw their inspiration.

## Methods
### Materials
Acrylamide was purchased from MP Biomedicals LLC (France, purity ≥ 99%, acrylamide acid <0.001%), lithium bromide (≥99%), sodium bromide (≥99%), sodium bicarbonate (≥99.5%), methacrylate anhydride (99%), N,N′-Methylenebisacrylamide (MBAA, purity ≥ 99.5%), microcrystalline cellulose (MCC, size exclusion, ca 50 μm particle size), 2,2,6,6-tetramethylpiperidine 1-oxyl (TEMPO, 98%), phenylboronate acrylamide (PBAAm, 98%), phosphotungstic acid hydrate (PTA, ≤ 0.002% total nitrogen), 2,4,6-trimethylbenzoyl chloride (97%) were bought from Sigam-Aldrich (Germany). Dimethoxyphenylphosphine (97%), dopamine hydrochloride (99%) and 2-butanone (99%) were received from Alfa Aesar (Germany). Sodium hydroxide (99%), borax-NaOH buffer (pH = 10.00, accuracy: ± 0.02 pH, reference temperature 20 °C) and sodium hypochlorite solution (14%, active chlorine) were purchased from TH Geyer (Germany). DI water (conductivity <1 μS/cm, room temperature) was used in all steps and all solvents were used directly from TH Geyer (Germany) without further treatment. Dopamine methylacrylate (DMA) and lithium phenyl-2,4,6-trimethylbenzoylphosphinate (LAP) were prepared using the reported method[54,55].

### Synthesis of DMA
Typically, 10 g of sodium and 4 g of NaHCO₃ were dissolved in 100 mL of DI water, bubbling with Argon for 15 min. Then, 5 g of dopamine-HCl was added, followed by the dropwise addition of 4.7 mL of methacrylate anhydride in 25 mL of THF, during which the pH was kept above 8 via adding NaOH (aq) as necessary. The reaction was stirred overnight at room temperature under Argon atmosphere. The aqueous mixture was washed twice with 100 mL of ethyl acetate. Then the pH of the solution was reduced to below 2 and extracted with 100 mL of ethyl acetate. The final ethyl acetate layers was dried over MgSO₄, then 300 mL of hexane was added with vigorous stirring and the suspension was stored at 4 °C overnight. The product was recrystallized from hexane and dried to yield 1.7 g of grey power. ¹H-NMR spectroscopy (400 MHz, DMSO-$d$): δ 6.62-6.58 (m, 2H), 6.42 (d, 1H), 5.62 (s, 1H), 5.30 (s, 1H), 3.27-3.20 (m, 2H), 2.56 (t, 2H), 1.84 (s, 3H). The peak assignments can be found in Supplementary Fig. 38. Shift of all peaks are consistent with expected values.

### Synthesis of LAP
At room temperature and under argon, 3.2 g (0.018 mol) of 2,4,6-trimethylbenzoyl chloride was added dropwise to 3.0 g of dimethyl phenylphosphonite with stirring. The reaction mixture was stirred for 18 h. Then a four-fold excess of lithium bromide in 100 mL of 2-butanone was added to the reaction mixture. Later, the mixture solution was heated to 50 °C. After 10 min, a solid precipitate formed. The mixture was cooled to ambient temperature, allowed to rest for 4 h and then filtered. The filtrate was washed and filtered three times

with 2-butanone to remove unreacted lithium bromide and excess solvent was removed by vacuum. The product obtained was a white solid. $^1$H- NMR spectroscopy (400 MHz, D$_2$O): δ 7.66 (m, 2H), 7.51 (m, 1H), 7.41 (m, 2H), 6.83 (s, 2H), 2.18 (s, 3H), 1.96 (s, 6H). The peak assignments can be found in Supplementary Fig. 39. Shift of all peaks are consistent with expected values.

## Isolation of CNCs

CNCs were synthesized via TEMPO-mediated oxidation with MCC as raw materials[56]. The CNC suspension in water was stored in refrigerator at 4 °C before use.

## Preparation of CNC hybrid hydrogels

The CNC hybrid hydrogels were prepared according to the reported work[23]. Usually, the precursor of hydrogels consists of CNCs, 2 M acrylamide (142 mg/mL), 1.25 mol% PBAAm/DMA (based on acrylamide) and solvents. The solvents were prepared as (1) a binary solvent system of borax-NaOH buffer solution containing 10 *vol*% ethanol, named 10% ethanol, (2) borax-NaOH buffer solution at pH 10 and (3) borax-NaOH buffer solution containing 5 wt% glucose dissolved in borax-NaOH buffer solution, named 5% glucose. The CNC suspensions were prepared first by redispersion in these solvents under sonication for about 3 min before the other components were added. The final concentration of CNCs in the precursor is 2 wt%. The ratio between monomer and CNCs was about 7:1. After adding 0.5 wt% LAP into the hydrogel precursor solutions under sonication, photo-initiated radical polymerization was carried out with a 15 W UV light (λ = 356 nm, irradiance around the precursor solution is 1.8 mW/cm$^2$) for 30 min, after the precursors with LAP were transferred into the mold. A cylindrical plastic tube with an internal diameter of 4.60 mm was used as the mold for shaping the hydrogels. The covalently crosslinked hybrid hydrogels with similar crosslinked density ( ~ 3.85 mg/mL) were prepared by using MBAA as crosslinker, and the other compositions were kept the same.

## Preparation of xerogel fibers

The xerogel fibers were prepared via uniaxial stretching (Z3 micro tensile test machine, Grip-Engineering Thümler GmbH, Germany) of cylinder-shaped hybrid hydrogels and following air drying in constant conditions (illustrated in Supplementary Fig. 5). A cylinder cavity with ca. 4.3 mm of inner diameter was used as the model so as to fix the initial shape of hydrogels. The elongation ratio was set as 15 (deformation ratio 1500%). The crosshead speed was 6 mm/min, and the initial strain rate was 0.025 s$^{-1}$.

## Simulation of water evaporation in porous materials

A COMSOL Multiphysics® software was utilized to simulate the evaporation process of water in porous materials. The parameters, including porosity, permeability, thermal conductivity, and matrix heat capacity, were set according to the literatures[57–61] (Supplementary Table 1). The density of the porous matrix was established at 1200 kg/m$^3$.

## Dynamic water sorption (DVS)

A DVS apparatus (DVS Advantage, surface measurement systems, London, UK) was used to evaluate the dynamic water vapor sorption behaviors of xerogel fibers. The xerogel fibers originating from various solvents with various lengths and weights of several milligrams were used for measurements. All measurements were carried out at constant temperature of 25 °C and a nitrogen flow of 200 sccm (standard cubic centimeter at 0 °C, 100 kPa per minute according to IUPAC). BJH method was applied to calculate the Pore Size Distribution[62].

Two procedures were applied. First, xerogel fibers were dried at 0% RH before the RH was increased to 80% and afterward decreased to

0% (Supplementary Fig. 29). In the next step a typical isothermal sorption-desorption process was executed (Supplementary Fig. 34). First, the samples were dried at 0% RH before the RH was increased stepwise in the following sequence: 5, 15, 25, 35, 45, 55, 65, 75, 85, and 95% RH (absorption curve), which was followed by a decrease to 0% RH in the reverse order (scanning desorption curve). During this procedure, the sample was regarded as saturated under a certain RH, thus reaching its equilibrium state, when the mass change per min (*dm/dt*) was lower than 0.001% min$^{-1}$ over a period of 10 min. The moisture content at each RH was calculated on the basis of the initial dehydrated mass of the xerogel fibers.

## Characterization

The irradiance of UV light was determined by sliver line UV-radiometer (CON-TROL CURE® UV SENSORS, Germany). The mechanical properties of the materials were recorded using a Z3 micro tensile test machine, which was equipped with a 50 N load cell. Young's modulus of hydrogels was calculated from strain-stress curves in the strain range from 0% to 100%. DMTA measurements were carried out using a DMA GABO EPLEXOR system (NETZSCH GABO Instruments GmbH). The measurements were conducted with a contact force of 0.2 N, a static strain of 0.5 ± 0.1% and a dynamic strain of 0.1 ± 0.05%. The rheological properties of hybrid hydrogels were tested using a HR 20 rheometer (TA Instruments, USA) with the UV accessory (OmniCure, Germany). A parallel plate with 20.00 mm diameter was applied and the gap was set to 300 μm. Humidity sweep tests were carried out with RH increasing rate of 2% min$^{-1}$ at 25 °C. The geometric shape and dimensions of CNCs were determined employing a CM 12 transmission electron microscope (Philips, the Netherlands) with the help of a negative stain method (2 wt% Phosphotungstic acid). A LEO supra-35 high-resolution field emission scanning electron microscope (Carl Zeiss AG, Germany) was used to characterize the microstructure of various samples and the targeted voltage was 5 kV. The oblique cutting xerogel samples were prepared by manual operation with the help of a stereomicroscope. The vertical cutting xerogel samples were produced using the Leica HistoCore BIOCUT system. The alignment structures of xerogel fibers were demonstrated under crossed polarization with an Eclipse 600 microscope from Nikon. An LV100 microscope with Berek compensator from Nikon was used to calculate the birefringence of materials. The number of carboxyl groups was measured by conductometric titration of the CNC suspensions[63].

## Data availability

The data generated in this study are provided in the Supplementary Information/Source Data file. All other relevant data supporting the findings of this study are available from the corresponding authors upon request. Source data are provided with this paper.

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

## Acknowledgements

K.Z. thanks the German Research Foundation (DFG) and Lower Saxony Ministry of Science and Culture for the project INST186/1281-1/FUGG. D.X. thanks the China Scholarship Council (CSC) for the financial support. We thank Prof. Dr. Marcus Müller from the Institute for Theoretical Physics, Georg-August-University of Göttingen, for valuable suggestions and discussion. We thank Dr. Kristian Hantke from the Max Planck Institute for Dynamics and Self-Organization for providing technical support in measuring the birefringence values. Mr. Siyuan Liu is acknowledged for the comments on the figures.

## Author contributions

K.Z. developed the concept. D.X. performed most of the experiments, analyzed the experimental data and wrote the paper. Y.Y. supported the preparation and analysis of hydrogels and their drying process. L.E. helped to perform DVS measurements and analyzed the DVS data. D.X., Y.W. and K.Z. discussed the simulation. D.X. ran the simulation, and Y.W. supported the simulation by improving the physical models. All authors discussed the results and revised the manuscript.

## Funding

## Competing interests

The authors declare no competing interests.
