## [Peer Review File · Nature Communications]

Divergent Deborah number-dependent transition from homogeneity to heterogeneityEditorial Note: Parts of this Peer Review File have been redacted as indicated to remove third-party material where no permission to publish could be obtained.

REVIEWER COMMENTS

Reviewer #1 (Remarks to the Author):

The manuscript by Xu et al. touches on a very interesting topic, namely how processing time in dynamic polymer materials influence structural heterogeneity in the materials. The topic is certainly of broad interest and worth studying in depth; however, the present manuscript requires some work before it should be considered for publication.

1. The central argument of the authors is that the experimental processing time, controlled by evaporation, relative to the relaxation time of the material, characterized by the dimensionless number the Deborah number, controls the transition from heterogeneity to homogeneity in the materials. However, the central conceit relies on a highly anisotropic process of evaporation, where local hydrodynamic timescales are much faster than they refer to. In order to make such broad and sweeping claims about the role of De , the authors need to demonstrate this phenomenon more generally and for systems where the anisotropy observed is not expected from the anisotropy of the process. For example, the evaporation produces structures that would be expected based on the radial phenomenon. What would happen for example in different geometries with similar compositions? What about processing the materials in different ways where one can control De , for example, by extrusion? These points should be discussed or the very broad conclusions should be softened substantially.

2. The schematic in Figure 3d and as used in the TOC is rather confusing. The authors argue that a single dimensionless parameter controls heterogeneity in the sample but yet the sample, i.e., 10% ethanol has regions with De above and below 1. This should be represented more clearly and explained in simpler language.

3. In general, the writing of the manuscript makes the conceptual understanding more

convoluted than clear. The authors should spend some time to rewrite the manuscript seeking simple and clear explanations for the observed phenomenon. The topic is certainly interesting and there is something in the current work that seems like it can be distilled and supported by the data, but the current approach makes it very opaque for the reader and only focuses on high-level or broad claims that are not supported by the data.

Reviewer #2 (Remarks to the Author):

Overall, I thought this was an interesting article, and the concept of Deborah number-dependent transition on the gelation, the resulting structure and resulting properties of CNC-polymer hydrogels was novel, and the results are noteworthy. However, there are several issues with this manuscript that need to be addressed.

The following aspect must be rectified in this paper.

1. Remove all connection to plants/trees mechanisms and bioinspired structure/design/function references in this paper (e.g., in the text, figures, and conclusions). The hierarchical structure in plants and trees, their biosynthesis process, and the resulting interaction with environment, are so complex, there is no connection to the trivial structures, processing and environmental response described in this study. It is fine that the authors are “inspired” by nature, but how the authors try to link this work back to biosynthesis processes is meaningless and undermines the quality and utility of this work from a knowledge and engineering perspective.

2. The authors need to provide how this work is different from prior work in CNC-polymer hydrogel systems. There is a lot of research in the cellulose nanomaterial community regarding self-assembly, gelation, and stimulus response. I still believe the concept of Deborah number that the authors use here is new, but it needs to be put into proper context. To start with the authors should consider looking at the following papers.

o De France, K. J., Hoare, T., & Cranston, E. D. (2017). Review of hydrogels and aerogels containing nanocellulose. *Chemistry of Materials*, 29(11), 4609-4631.

o Babaei-Ghazvini, A., & Acharya, B. (2022). Influence of cellulose nanocrystal aspect ratio on shear force aligned films: Physical and mechanical properties. *Carbohydrate Polymer*

Technologies and Applications, 3, 100217.

o Li, K., Clarkson, C. M., Wang, L., Liu, Y., Lamm, M., Pang, Z., ... & Ozcan, S. (2021).

Alignment of cellulose nanofibers: harnessing nanoscale properties to macroscale benefits.

ACS nano, 15(3), 3646-3673.

o Mittal, N., Ansari, F., Gowda. V, K., Brouzet, C., Chen, P., Larsson, P. T., ... & Soderberg, L.

D. (2018). Multiscale control of nanocellulose assembly: transferring remarkable nanoscale fibril mechanics to macroscale fibers. ACS nano, 12(7), 6378-6388.

o Mendez, J., Annamalai, P. K., Eichhorn, S. J., Rusli, R., Rowan, S. J., Foster, E. J., & Weder, C.

(2011). Bioinspired mechanically adaptive polymer nanocomposites with water-activated shape-memory effect. Macromolecules, 44(17), 6827-6835.

o Annamalai, P. K., Dagnon, K. L., Monemian, S., Foster, E. J., Rowan, S. J., & Weder, C.

(2014). Water-Responsive Mechanically Adaptive Nanocomposites Based on Styrene–Butadiene Rubber and Cellulose Nanocrystals - Processing Matters. ACS applied materials & interfaces, 6(2), 967-976.

3. Need to give TEM image to show the morphology of CNC used in this study. Not all CNC are the same, they are highlight depended on their processing. In the supporting info document the authors should provide some TEM images of their CNCs.

4. The authors need to give greater details on the approached used to measure the CNCs and how many CNCs were measured. Since the CNC gelation and self-assembly characteristics are dependent on CNC morphology it is important to demonstrate that the reported dimensions of the CNCs used in this work are based off of sound measurement practices. See following article on how such measurement should be made, and the recommendation that at least 300 CNCs should be measured.

o Meija, J., Bushell, M., Couillard, M., Beck, S., Bonevich, J., Cui, K., ... & Johnston, L. J.

(2020). Particle size distributions for cellulose nanocrystals measured by transmission electron microscopy: an interlaboratory comparison. Analytical chemistry, 92(19), 13434-13442.

5. In Section "preparation of CNC hybrid hydrogels". More information is needed on the hydrogel was prepared in this study. How were CNCs added to the solvents (mixing,

ultrasound)? What was the CNC wt% in the hydrogel solution? What was the ratio of CNC to polymer in the hydrogel solution? What was the dimensions of the as cast hydrogel (before stretching)?

6. In the supporting information document, the authors should provide a series of images of the fiber cross-section to show and highlight the features that they used to indicate the “outer layer”. From the images given in the paper in Figure 2b, 3e, and 4e it is nearly impossible to see the features that the authors claim that are there. In by doing this, it would greatly increase the credibility of what the authors are claiming in the structural feature changes within these fibers. The current images of fiber cross sections Figures S3, S6, S8 and S15 in the supporting document are also ambiguous as to highlighting the outer layer. For S15, the authors add a dashed line, so this “masks” the reader from seeing the interface between outer and inner layer. The authors should minimize such overlay as it biases what the reader sees.

7. Likewise, for Figure 2: These series of SEM images in part b, it is unclear what they are showing, the features and differences in features that the authors claim are not seen. This also puts into question to how the authors measured the thickness of the outer layer. and puts into question the T/R information

8. Did the authors independently correlate Birefringence to CNC alignment, such as with x-ray diffraction, etc. ? What other factors influence birefringence of CNC-polymer systems? Such information should be included in the manuscript.

9. The authors should assess the structural differences and changes of outer vs inner layers. The outer layer was denser, but was the density of the inner and outer layers measured? In Line 387-388 Lower water transmission rate of outer layer, indicates something is different, which could be a combination of several factors, such as differences in pore size and density of this outer layer. The SEM image in Figure S3, shows the pore structure. It would be good if the authors could use similar technique to look had how the pore structure changes for each of the samples, with and without stretching.

10. Authors should estimate the three evaporation speeds, low, medium, and high is too ambiguous.

11. In general I found that the montage of schematics and images within Figures 1,2,3, and 4 difficult to "see". They schematics and images are too small to see the meaningful information that the authors refer to in the text. Thus in makes these figures in effective in sharing information.

12. In the conclusion the authors state "Our work verified the inner connection between soft matters and living things as well". This work did not show any results that link these two aspects. As I mentioned in issue #1.... the authors should remove all

Other aspect the authors should consider addressing.

13. line 322-323. should be more specific as to "relatively" solid vs liquid. If the hydrogel was a liquid, then how can measure mechanical properties, how could it be stretched.

14. Lines 348 to 353. It is unclear how this hydrogel is different from the one described in lines 339 to 347.

15. line 499.... unclear what deformation speed is and this does not seem to match the units of (1/s).

One Aspect I liked: The supporting document had a lot of information, figures and table... this was wonderful and helped me greatly in understanding this work.

Reviewer #1 (Remarks to the Author):

The manuscript by Xu et al. touches on a very interesting topic, namely how processing time in dynamic polymer materials influence structural heterogeneity in the materials. The topic is certainly of broad interest and worth studying in depth; however, the present manuscript requires some work before it should be considered for publication.

Response: We thank the reviewer for the very positive comments on our research work. We are pleased to learn that the reviewer fully recognized the importance and novelty of this study. Below, we have provided a point-by-point response to the valuable comments.

1. The central argument of the authors is that the experimental processing time, controlled by evaporation, relative to the relaxation time of the material, characterized by the dimensionless number the Deborah number, controls the transition from heterogeneity to homogeneity in the materials. However, the central conceit relies on a highly anisotropic process of evaporation, where local hydrodynamic timescales are much faster than they refer to. In order to make such broad and sweeping claims about the role of De , the authors need to demonstrate this phenomenon more generally and for systems where the anisotropy observed is not expected from the anisotropy of the process. For example, the evaporation produces structures that would be expected based on the radial phenomenon. What would happen for example in different geometries with similar compositions? What about processing the materials in different ways where one can control De , for example, by extrusion? These points should be discussed or the very broad conclusions should be softened substantially.

Response: We thank the reviewer for the valuable comments. As the reviewer mentioned, the evaporation process is one of critical parameter in forming heterogeneous structures. We have tried our best to eliminate the geometry-induced anisotropic air drying process. This is to present reliable and repeatable research work.

In view of the geometry parameters of the specimens, the ideal uniform shape of the specimens could be spherical. However, the evaporation process for hydrogels, even for spherical hydrogels, involves a competition between water transport within polymer matrix and water transport away

from the polymer matrix. A core shell structure has been reported in swelling process (reverse process of dehydration) when the gel bead was applied (*Phys. Rev. Appl.*, **2016**, 6(6): 064010). On the other hand, due to the viscoelasticity of dynamic hydrogels (Supplementary Fig. 5), it can deform if the drying time is too long due to its own gravity. The deformation will be obvious around the contact line **and induces the “artifacts”** of heterogeneity. More importantly, based on the theoretical analysis, it has been shown that the air drying process is almost isotropic for cylindrical samples, as it is for spherical samples (Supplementary Fig. 7). The main air drying was carried out along with the radial axis and the water transition along the longitudinal axis can be ignored for the central zone of cylindrical samples.

[FIGURE REDACTED]

Fig. Core shell structure found in the swelling process of bead gel. (*Phys. Rev. Appl.*, **2016**, 6(6): 064010)

Supplementary Fig. 5. The rheological properties of dynamically crosslinked hydrogel. a) Frequency sweep; b) Shear thinning properties.

Supplementary Fig. 7. Based on numerical analysis, it was demonstrated that the air drying process is isotropic for cylindrical samples. The detailed discussion has been presented in Supplementary Discussion.

Herein, we use the cylindrical hydrogels, which represent 1D structures. By using uniaxial stretching, the diameter of the hydrogels was reduced to about 1 mm, allowing the dehydration process to be carried out very quickly. At the same time, it avoids local deformation due to gravity and viscoelasticity, and introduces homogenous air drying process for hydrogel samples. Considering the possible radial phenomenon, we set three groups of hydrogels, which are of the same composition and the only difference is solvents. By comparison it was demonstrated that the anisotropy was dominated by the differentiation of De . Specifically, as for the 10% ethanol and buffer samples, the De behaved larger than 1 and less than 1 for different regions. As the result, obtained xerogels showed anisotropic structures. However, for the 5% glucose samples the obtained xerogels showed the isotropic structures, while the geometric parameters of the starting hydrogels were the same as the other two groups. Therefore, the anisotropy of xerogels should mainly be due to the divergence of De rather than radial phenomenon.

Moreover,

1. The covalently crosslinked hydrogels were introduced as control group. Typically, the relaxation time (τ) of covalently crosslinked hydrogels is of the order of 10^{-5} s, and the relaxation behaviour is attributed to the concentration fluctuations of the polymer chains, which are present in most hydrogels (*Polym. Chem.*, **2019**, 10(25): 3503-3513). Therefore, for both central regions and

boundary regions the De are of low values ($\ll 1$, due to the ratio between relaxation time and interaction time). For such hydrogels, there is no divergence of De and, as expected, a similar microstructure was found in whole samples, while the radial phenomenon was not observed (Supplementary Fig. 26).

Supplementary Fig. 26. The microstructures of the xerogel obtained from covalently crosslinked hydrogels. A cylindrical hydrogel was applied before air drying. a) Central region, b) Boundary region. The relaxation time of such covalently crosslinked hydrogel is in the order of 10^{-5} s, which is assigned to the concentration fluctuations of the polymer chains in the covalent crosslinked hydrogels^{6,7}. Thus, the De for central or boundary region was approached to zero. Both the boundary zone and the central zone showed almost the same structural characteristics according to the SEM images.

2. Xerogels originating from hydrogels of different shapes were analyzed. As shown in Supplementary Fig. 23, when a cubical hydrogel was used to form xerogel, the anisotropy was also introduced because of the geometric asymmetry. Usually, the evaporation speed around the corners/edges will be completely different from the central areas of the faces. Because of the asymmetric air drying process in these areas, the heterogeneous structures around the corners/edges were widely found (Supplementary Fig. 23, 24). Besides, similar anisotropic structures can also be observed in the face zones of cuboidal xerogel samples of sufficiently high thickness. In comparison, when the thickness of cubical hydrogel was decreased to very low values, which means the evaporation process can be finished within a short time to eliminate the difference between the inner / outer regions, the heterogeneity was not detected (Supplementary Fig. 25). In this situation, both central region and boundary regions have the De of high values (> 1) without divergence of De . This result was also in good agreement with the experimental expectation.

Supplementary Fig. 22. The anisotropic evaporation process for cuboid hydrogels. a). Illustration of the evaporation speed distribution around a cuboid hydrogel sample surface. b-d). The relative humidity distribution around the cubic hydrogel sample after 90 s air drying. A cuboid sample was rotated in certain directions to simulate difference in air drying process due to geometric anisotropy (0° , 30° , 45°). Due to anisotropy of the geometric parameters, the evaporation behavior will be intrinsically anisotropy as well.

Supplementary Fig. 11. The microstructures of xerogel fibers with rectangular cross section. The heterogeneous structures were distributed around the boundary zone. In general, the inner part was rough with cracks while the outer part was relatively smooth.

Supplementary Fig. 23. The microstructures of obtained xerogel after applying an external force. The force was applied for about 5 minutes to allow the network reorganization. Then the force was removed. During the evaporation process, one side was placed on the glass and the other side was open. The heterogeneous structure emerged around the corner due to the air drying process, and the heterogeneity was also found in the face region for sufficiently high thickness. This is ascribed to divergent De .

Supplementary Fig. 24. The microstructures of obtained xerogel after applying an external force. The force was removed before air drying. During the evaporation process, both sides were covered with glass slides. The only heterogeneous structure was found around the corner of xerogels. In contrast, the central zone showed the same microstructural features, and no obvious heterogeneity was detected. By numerical simulation it can be found that different evaporation processes induced by the anisotropic geometry are obvious. Dehydration at the edge of the samples is much faster and more anisotropic than in the center. In the center of the samples, the evaporation speed was the same, i.e., those regions are of the same De (approx. 0.054, based on a rough model). The same De without divergence predicted the homogeneity of the resulting xerogel samples well.

Supplementary Fig. 25. The extremely thin xerogel obtained by using dynamically crosslinked hydrogels after air drying. a) SEM image showing the microstructure of xerogel films; b) The residual water content in air drying process. Due to the rather small thickness and the large evaporation area, the De for both the boundary and the central region exceed 1 without divergence and as expected the heterogeneous structures are not detected.

3. The heterogeneity was not observed in the virgin hydrogels or in the early stage of air drying, indicating that the heterogeneity was due to the dynamic air drying process rather than the intrinsic heterogeneity of the samples (Supplementary Fig. 6 and Fig 3e).

Supplementary Fig. 6. SEM images of virgin CNC hybrid hydrogels (without stretching and air drying) after freeze-drying. a) inner part; b) outer part. Scale bar: 10 μ m.

Fig 3e. RH distribution around hydrogel fibers at various times and corresponding SEM images. Scale bars: 10 μ m.

Based on these experimental data, we believe that the evolution of the system from homogeneity to heterogeneity was dominated by the divergence of De , excluding the asymmetry of the process.

Corresponding description was amended in the manuscript on Page 6 and 15:

“It is worth emphasizing that the evaporation process in such cylindrical samples is the isotropic process along the radial direction, thus avoiding the intrinsic heterogeneity induced by the radial phenomenon (more details in Supplementary Fig. 7).”

“The transition from homogeneity to heterogeneity in air drying process of hydrogels was also studied under different conditions on samples of various shapes (Supplementary Fig. 22-26). The homogeneity and heterogeneity in the resulting xerogels also follow the divergent De induced structural transitions.”

Considering the rheological properties (Supplementary Fig. 5) of these dynamically crosslinked hydrogels under other processing conditions, it would be difficult to stabilize the shape by extrusion. During extrusion, the external force was applied to simulate the manipulation of the external field induced diverse De , and can induce partial organization of the polymer network. Due to the dynamic nature of the polymer network, the polymer network will be reorganized once the external force is removed. Thus, in general we can imagine that once the hydrogels have been extruded from the nozzle, the outer part could retain the alignment structure due to shearing and fast dehydration, while the inner part will become lower order structures after relaxation.

2. The schematic in Figure 3d and as used in the TOC is rather confusing. The authors argue that a single dimensionless parameter controls heterogeneity in the sample but yet the sample, i.e., 10% ethanol has regions with De above and below 1. This should be represented more clearly and explained in simpler language.

Response: We appreciate the reviewer's suggestion and insightful comments. We apologize for any misleading descriptions or information due to improper demonstration. We further improved Fig 3 and TOC to demonstrate this result more clearly.

The dimensionless parameter De does play the important role in describing such air drying process, but the transition from homogeneity to heterogeneity originates from the divergence of De . The critical point of De is given by $De=1$. It is generally considered that the material is relatively solid-like with De more than 1; and the material is relatively liquid-like if De is lower than 1. During drying, the samples will gradually have different regions. For instance, the samples containing 10% ethanol has regions with De above (outer part) and below (inner part) 1. As a result, the sample contains different microstructures in the two regions resulting in the formation of anisotropic structures. More results based on samples or porous matrix of different shapes are also discussed, whose experimental data also support the conclusion. By this, we hope that we have precisely demonstrated the finding of this study as that the divergence of De can induce the transition from homogeneity to heterogeneity.

Fig 3. Hydrogel-xerogel transition process during air drying.

TOC has also been revised to emphasize the novelty:

3. In general, the writing of the manuscript makes the conceptual understanding more convoluted than clear. The authors should spend some time to rewrite the manuscript seeking simple and clear explanations for the observed phenomenon. The topic is certainly interesting and there is something in the current work that seems like it can be distilled and supported by the data, but the current approach makes it very opaque for the reader and only focuses on high-level or broad claims that are not supported by the data.

Response: Thanks for the kind comments and suggestions. We hope through this revision that we could better highlight the innovation of this study and improve the readability of the manuscript, by better delivering and explaining these experimental data as well as by further work.

1. We have reorganized the figures and text in the manuscript, and the new finding focuses on the divergent De dependent transition from homogeneity to heterogeneity.

2. The manuscript has been carefully revised. The irrelevant text has been removed. The grammar and spelling have been carefully checked.

3. Additional experimental data have been provided to emphasize the innovation of the research work. More detailed information has been added to the manuscript.

Reviewer #2 (Remarks to the Author):

Overall, I thought this was an interesting article, and the concept of Dehorah number-dependent transition on the gelation, the resulting structure and resulting properties of CNC-polymer hydrogels was novel, and the results are noteworthy. However, there are several issues with this manuscript that need to be addressed.

Response: We appreciate the insightful and constructive comments and advice provided by the reviewer. We have carefully considered these concerns and tried our best to address the comments. Below, we have provided a point-by-point response to the valuable comments.

The following aspect must be rectified in this paper.

1. Remove all connection to plants/trees mechanisms and bioinspired structure/design/function references in this paper (e.g., in the text, figures, and conclusions). The hierarchical structure in plants and trees, their biosynthesis process, and the resulting interaction with environment, are so complex, there is no connection to the trivial structures, processing and environmental response described in this study. It is fine that the authors are “inspired” by nature, but how the authors try to link this work back to biosynthesis processes is meaningless and undermines the quality and utility of this work from a knowledge and engineering perspective.

Response: Thanks for the comments. We do understand the reviewer’s concerns about the massive difference between living things and soft materials. As we discussed in the manuscript (also have been widely reported, e.g., *Science* 363(6426): 504-508(2019)), the similarity between soft matters and the organism has been demonstrated. A critical field about biomaterials using hydrogels have been developed and still acts as the hot point (e.g., *Science* 364(6439): 458-464 (2019)). Thus, what we have done in this research work may provide a new angle for understanding such heterogeneous structures in nature, rather than to give any clear answer about such complex biosynthesis process.

Regarding the reviewer’s concern, we have carefully modified the manuscript to avoid any misleading information or deceptive relationship between biosynthesis process/ this study.

a). A major revision has been made to the manuscript. Fig. 1 has been modified. Less relevant descriptions have been removed and the topic has become more precise and focused, by only showing the heterogeneous structures.

Removed descriptions/text include but are not limited to the following:

“As the result of environmental adaption during modified air drying processes of hybrid hydrogels...”

“The growth of plants on the macroscopic level can be simplified as a “pseudo” uniaxial stretching process. In terms of cell growth, the cell wall will expand due to the internal pressure of the cell, resulting in the orientation of cellulose microfibers from perpendicular to parallel with respect to the cell axis...”

“...and then the adaptation process to...”

“Based on this heterogeneity of xerogel fibers, a similar transition similar as natural phenomenon should take place to generate these different layers. In nature, the plant morphology shows different structural features as the adapting response of organisms to various environmental conditions, e.g. the thickness of cuticle layer usually increases due to the increase of relative evaporation speed. The cactus is famous for the waxy layer on the surface of the stem, which does not appear in ferns and grapes.”

“is similar as the adaptation behaviors of plants by forming protective surface layers in nature.” “... reassembling the adaptive response of some plants to extreme surroundings in nature.” “The environmental adaptation behaviors are observed in the formation of xerogel fibers, e.g., the residual water content was found to be related to the existence of the outer layer, which is consistent with the behaviors of natural plants.”

“Our work verified the inner connection between soft matters and living things as well.”

b). Fig. 2 has been revised. We removed the description about the adaptation behaviors of plants in various environmental conditions. Only the structures found in xerogels were discussed.

c). The conclusion has also been modified, and the description was only based on the experimental data or direct inference.

Fig 1. Biomimetic heterogeneity generated in the transition process from hydrogel to xerogel.

Fig 2. Transition of xerogel fibers from homogeneity to heterogeneity in various mediums.

2. The authors need to provide how this work is different from prior work in CNC-polymer hydrogel systems. There is a lot of research in the cellulose nanomaterial community regarding self-assembly, gelation, and stimulus response. I still believe the concept of Dehorah number that the authors use here is new, but it needs to be put into proper context. To start with the authors should consider looking at the following papers.

- De France, K. J., Hoare, T., & Cranston, E. D. (2017). Review of hydrogels and aerogels containing nanocellulose. *Chemistry of Materials*, 29(11), 4609-4631.
- Babaei-Ghazvini, A., & Acharya, B. (2022). Influence of cellulose nanocrystal aspect ratio on shear force aligned films: Physical and mechanical properties. *Carbohydrate Polymer Technologies and Applications*, 3, 100217.

- Li, K., Clarkson, C. M., Wang, L., Liu, Y., Lamm, M., Pang, Z., ... & Ozcan, S. (2021). Alignment of cellulose nanofibers: harnessing nanoscale properties to macroscale benefits. *ACS nano*, 15(3), 3646-3673.
- Mittal, N., Ansari, F., Gowda. V, K., Brouzet, C., Chen, P., Larsson, P. T., ... & Soderberg, L. D. (2018). Multiscale control of nanocellulose assembly: transferring remarkable nanoscale fibril mechanics to macroscale fibers. *ACS nano*, 12(7), 6378-6388.
- Mendez, J., Annamalai, P. K., Eichhorn, S. J., Rusli, R., Rowan, S. J., Foster, E. J., & Weder, C. (2011). Bioinspired mechanically adaptive polymer nanocomposites with water-activated shape-memory effect. *Macromolecules*, 44(17), 6827-6835.
- Annamalai, P. K., Dagnon, K. L., Monemian, S., Foster, E. J., Rowan, S. J., & Weder, C. (2014). Water-Responsive Mechanically Adaptive Nanocomposites Based on Styrene–Butadiene Rubber and Cellulose Nanocrystals - Processing Matters. *ACS applied materials & interfaces*, 6(2), 967-976.

Response: Thanks for attentive advice. In fact, all papers mentioned by the reviewer are already cited in our manuscript as.

- De France, K. J., Hoare, T., & Cranston, E. D. (2017). Review of hydrogels and aerogels containing nanocellulose. *Chemistry of Materials*, 29(11), 4609-4631.
16. De France KJ, Hoare T, Cranston ED. Review of hydrogels and aerogels containing nanocellulose. *Chem. Mater.* 29, 46 09- 46 31 (2017).
 - Mendez, J., Annamalai, P. K., Eichhorn, S. J., Rusli, R., Rowan, S. J., Foster, E. J., & Weder, C. (2011). Bioinspired mechanically adaptive polymer nanocomposites with water-activated shape-memory effect. *Macromolecules*, 44(17), 6827-6835.
 7. Mendez J, *et al.* Bioinspired mechanically adaptive polymer nanocomposites with water-activated shape-memory effect. *Macromolecules* **44**, 68 27-68 35 (2011).
 - Annamalai, P. K., Dagnon, K. L., Monemian, S., Foster, E. J., Rowan, S. J., & Weder, C. (2014). Water-Responsive Mechanically Adaptive Nanocomposites Based on Styrene–Butadiene Rubber and Cellulose Nanocrystals - Processing Matters. *ACS applied materials & interfaces*, 6(2), 967-976.
 8. Annamalai PK, Dagnon KL, Monemian , Foster EJ, Rowan SJ, Weder C. Water-Responsive Mechanically Adaptive Nanocomposites Based on Styrene–Butadiene Rubber and Cellulose Nanocrystals & Processing Matters. *ACS Appl Mater Interfaces* **6**, 967- 976 (2014).

- Li, K., Clarkson, C. M., Wang, L., Liu, Y., Lamm, M., Pang, Z., ... & Ozcan, S. (2021). Alignment of cellulose nanofibers: harnessing nanoscale properties to macroscale benefits. *ACS nano*, 15(3), 3646-3673.
- Li K, *et al.* Alignment of cellulose nanofibers: harnessing nanoscale properties to macroscale benefits. *ACS nano* **15**, 3646- 3673 (2021).
- Mittal, N., Ansari, F., Gowda, V, K., Brouzet, C., Chen, P., Larsson, P. T., ... & Soderberg, L. D. (2018). Multiscale control of nanocellulose assembly: transferring remarkable nanoscale fibril mechanics to macroscale fibers. *ACS nano*, 12(7), 6378-6388.
- Mittal N, *et al.* Multiscale control of nanocellulose assembly: transferring remarkable nanoscale fibril mechanics to macroscale fibers. *ACS nano* **12**, 6378-6388 (2018).
- Babaei-Ghazvini, A., & Acharya, B. (2022). Influence of cellulose nanocrystal aspect ratio on shear force aligned films: Physical and mechanical properties. *Carbohydrate Polymer Technologies and Applications*, 3, 100217.
- Babaei-Ghazvini A, Acharya B. Influence of cellulose nanocrystal aspect ratio on shear force aligned films: Physical and mechanical properties. *Carbohydr. Polym. Technol. Appl.* **3**, 100217 (2022).

Following kind suggestions by reviewer #2, we improved the manuscript by describing the difference between our work and previous works in more detail as on Page 3:

“On the other hand, although nanocellulose has been widely used to improve the performance of composite materials due to its superior mechanical properties¹⁶ or to impart stimuli-responsive behaviors via its wetting properties^{17, 18}, or to develop the approaches^{19, 20} and strategies²¹ to realize the alignment of nanocellulose in bulk solutions or composite materials, the main focus of the current work lies on the structure formation and development of the materials depending on their intrinsic rheological properties. As well, this study also sheds light on understanding the effect of such a dynamic process, i.e., the dehydration of composites in the environment, on the resulting structures of materials.”

3. Need to give TEM image to show the morphology of CNC used in this study. Not all CNC are the same, they are highlight depended on their processing. In the supporting info document the authors should provide some TEM images of their CNCs.

Response: Thanks for the reviewer’s suggestive comments. We added more TEM images of the CNCs we used, including lower magnification and higher magnification (Supplementary Fig. 3).

The geometry parameters of the CNCs were statistically analyzed on the basis of 300 individual CNCs. The average length and width were 209 ± 55 nm and 14 ± 5 nm respectively, with an aspect ratio of ~ 15 .

The corresponding contents in manuscript on Page 3 was modified as below,
“Based on TEM images of 300 individual CNCs after TEMPO-mediated oxidation, CNCs of 209 ± 55 nm in length and 14 ± 5 nm in width were prepared, with an aspect ratio of ~ 15 (Supplementary Fig. 3)”

Supplementary Fig. 3. Typical TEM images of TEMPO-CNCs. The length and width were analyzed, and the number of samples for statistical analysis was 300.

4. The authors need to give greater details on the approached used to measure the CNCs and how many CNCs were measured. Since the CNC gelation and self-assembly characteristics are dependent on CNC morphology it is important to demonstrate that the reported dimensions of the CNCs used in this work are based off of sound measurement practices. See following article on

how such measurement should be made, and the recommendation that at least 300 CNCs should be measured.

- Meija, J., Bushell, M., Couillard, M., Beck, S., Bonevich, J., Cui, K., ... & Johnston, L. J. (2020). Particle size distributions for cellulose nanocrystals measured by transmission electron microscopy: an interlaboratory comparison. *Analytical chemistry*, 92(19), 13434-13442.

Response: Thanks for the suggestive comments. Using TEM image analysis, the geometry parameters of the CNCs we used were discussed in detail on the basis of 300 individual CNCs. One the whole, the shapes and sizes of CNCs were homogenous. The average length and width were 209 ± 55 nm and 14 ± 5 nm respectively, with an aspect ratio of ~ 15 . The corresponding contents in manuscript on Page 3 was modified as below,

“Based on TEM images of 300 individual CNCs after TEMPO-mediated oxidation, CNCs of 209 ± 55 nm in length and 14 ± 5 nm in width were prepared, with an aspect ratio of ~ 15 (Supplementary Fig. 3)”

5. In Section "preparation of CNC hybrid hydrogels". More information is needed on the hydrogel was prepared in this study. How were CNCs added to the solvents (mixing, ultrasound)? What was the CNC wt% in the hydrogel solution? What was the ratio of CNC to polymer in the hydrogel solution? What was the dimensions of the as cast hydrogel (before stretching)?

Response: We thank reviewer for the valuable comments. More details about “preparation of CNC hybrid hydrogels” have been added and the corresponding description in the manuscript has been revised as below,

“Usually, the precursor of hydrogels consists of CNCs, 2 M acrylamide (142 mg/mL), 1.25 mol% PBAAm/ DMA (based on acrylamide) and solvents.”

“The CNC suspensions were prepared first by redispersion in these solvents under sonication for about 3 min, before the other components were added. The final concentration of CNCs in the precursor is 2 wt%. The ratio between monomer and CNCs was about 7: 1. After adding 0.5 wt% LAP into the hydrogel precursor solutions under sonication, photoinitiated radical polymerization was carried out with a 15 W UV light ($\lambda = 356$ nm, irradiance around the precursor solution is 1.8 mW/cm²) for 30 min, after the precursors with LAP were transferred into the mold. A cylindrical plastic tube with an internal diameter of 4.60 mm was used as the mold for shaping the hydrogels.”

6. In the supporting information document, the authors should provide a series of images of the fiber cross-section to show and highlight the features that they used to indicate the “outer layer”. From the images giving in the paper in Figure 2b, 3e, and 4e it is nearly impossible to see the features that the authors claim that are there. In by doing this, it would greatly increase the credibility of what the authors are claiming in the structural feature changes within these fibers. The current images of fiber cross sections Figures S3, S6, S8 and S15 in the supporting document are also ambiguous as to highlighting the outer layer. For S15, the authors add a dashed line, so this “masks” the reader from seeing the interface between outer and inner layer. The authors should minimize such overlay as it biases what the reader sees.

Response: Thanks for the suggestive comments.

1. Fig 2, 3 and 4 have been modified for better clarity. Please also see the figures as follows.

Fig 2. Transition of xerogel fibers from homogeneity to heterogeneity in various mediums.

Fig 3. Hydrogel xerogel transition process during air drying.

Fig 4. Water vapor sorption/desorption behaviors of various xerogel fibers.

2. As for Supplementary Fig. 3 (in the revised manuscript as Supplementary Fig. 6), there was no outer layer. Related description is to be found in the manuscript as below,

“Homogeneous porous structure of freeze-dried hydrogels as shown in SEM image indicates the homogeneity of as-prepared hydrogels and thus uniform distribution of CNCs.”

“Interestingly, we found a prominent heterogeneous structure in the dried xerogel fibers, which is different than the homogeneous porous structures in virgin hydrogels”

“In contrast, as prepared hybrid hydrogels (without stretching and air drying) only contained the typical porous structure with homogenous distribution, with the same thickness of boundary layer and internal pore walls”
3. Supplementary Fig. 6, Supplementary Fig. 8, and Supplementary Fig. 15 (in the revised manuscript as Supplementary Fig. 9, Supplementary Fig. 12, Supplementary Fig. 21) have been modified. The contrast of SEM images was improved.

Supplementary Fig. 9. SEM image of the cross section of a xerogel fiber. The colored arrows indicate different regions of different structural features.

Supplementary Fig. 12. The optical microscopy images, polarized microscopy images and SEM images of various obtained xerogel fibers. The colored arrows indicate the regions with different structural features.

Supplementary Fig. 21. The heterogeneous structures were found in obtained xerogel fibers from buffer samples without stretching. The virgin hydrogels were placed vertically or horizontally. Scale bar: 10 μm. Insets: the original status of hydrogels.

7. Likewise, for Figure 2: These series of SEM images in part b, it is unclear what they are showing, the features and differences in features that the authors claim are not seen. This also puts into question to how the authors measured the thickness of the outer layer. and puts into question the T/R information

Response: Thanks for the suggestive comments. Figure 2 has been modified as mentioned above. The detailed information about T/R was provided as shown in Supplementary Fig. 13.

Supplementary Fig. 13. The microstructures of xerogel fibers with circular cross section. T: the thickness of outer part; R: the radius of xerogel samples.

8. Did the authors independently correlate Birefringence to CNC alignment, such as with x-ray diffraction, etc. ? What other factors influence birefringence of CNC-polymer systems? Such information should be included in the manuscript.

Response: We thank the reviewer for attentive question about the relationship between CNC alignment and birefringence. Besides the reported work by other researchers (ref 30, 31), our

group also independently characterized the birefringence and CNC alignment in our previous works with the same composition of hydrogels and the positive correlation relationship between CNC alignment and birefringence was concluded (ref 23). In detail, Wide-angle X-ray scattering (WAXS) of the xerogels was carried out at the Shanghai Synchrotron Radiation Facility (SSRF) on the BL16B beamline with an X-ray wavelength of 0.124 nm. Intensity distribution profiles in the azimuthal angle (ϕ) were used to calculate the orientation index (π) according to the equation:

$$\pi = \frac{180^\circ - fwhm}{180^\circ}$$

where *fwhm* is the full width of the half-maximum of the azimuthal profiles from the selected equatorial reflection.

[FIGURE REDACTED]

Fig. Birefringence at position No. 4 (P4) in and corresponding orientation indexes of CNCs. This result is reported in our previous work in ACS Nano 2019, 13, 4, 3867–3874, as can be found in reference [23].

The birefringence was dominated by the alignment of CNCs, because the reorganization of dynamic bonding could eliminate the alignment of polymer chains. The birefringence changes with variable CNC concentrations with the detailed information was shown as below,

[FIGURE REDACTED]

Fig. Birefringence of hybrid hydrogels with various CNC amounts. This result is reported in our previous work in ACS Nano 2019, 13, 4, 3867–3874, as can be found in refence [23].

We added more description and discussion in the revised manuscript on Page 6 and 10, *“CNCs within the polymer matrix are ordered during both the uniaxial stretching and subsequent air drying process²³, while the alignment of polymer chains was eliminated due to the fast cleavage and rebinding of dynamic bonds between borate and catechol moieties.”*

“Since the alignment of polymer chains can be eliminated by the reorganization, the birefringence of resulting materials was dominated by the orientation of CNCs.²³”

9. The authors should assess the structural differences and changes of outer vs inner layers. The outer layer was denser, but was the density of the inner and outer layers measured? In Line 387388 Lower water transmission rate of outer layer, indicates something is different, which could be a combination of several factors, such as differences in pore size and density of this outer layer. The SEM image in Figure S3, shows the pore structure. It would be good if the authors could use

similar technique to look had how the pore structure changes for each of the samples, with and without stretching.

Response: Thanks for the suggestive comments. As the reviewer addressed, the difference on water transmission rate was induced by the anisotropy of inner/outer part. It can be ascribed to the comprehensive effect of several factors.

1. The organization of polymers/CNCs has been discussed in the manuscript, e.g., as on Page 7 the first paragraph. Typically, the orientation degree of outer layer is higher than that of the inner layer.

2. The microstructures and the thickness of the outer layer and the inner layer were discussed directly by SEM image analysis. The outer layer was smooth and dense, while the inner layer was rough with cracks. The thickness of outer layer was evaluated as T/R values.

3. The pore size distribution of samples was conducted via BJH method. According to the PSD results, the primary and secondary pore diameters of xerogels (10% ethanol and buffer) with heterogeneous structures can be observed, which are centered at 2.6 nm and 8.1 nm. This result indicates the distinct pore structures for outer part and inner part. In contrast, the homogenous xerogels (5% glucose) shows that the primary, secondary and tertiary pore diameters are centered at 2.6 nm, 4.5 nm and 8.1 nm. In particular, the distributed frequency (refers to dV_p/dR_p) of the smallest pore is quite lower, which may explain the undetected outer layer in the 5% glucose xerogel fibers. The same largest pore diameter (centered at 8.1 nm) for all these samples indicates the similar porous structure in the inner parts (For the 5% glucose sample with no detected outer layer, it refers to almost the entire sample). The total pore volumes of different samples also indicate that the disappearance of the outer layer results in an increase in pore volume. From this, it can be concluded that the outer part was denser than the inner part. The density of outer layer and inner layer was also evaluated. As the result, the density of outer layer is about 1.23 g/cm^3 , while the density of inner layer is about 1.11 g/cm^3 . The difference in densities should be caused by the different pore structures and total pore volumes.

Based on above experimental data, we can understand the roles of outer layer and inner layer playing on water transition process. Such a dense, low pore volume outer layer with small pores, will dominate the water transmission process of the samples. However, it should be noted that the exact process of water transmission in porous structures, as an independent research field, is still very challenging. Therefore, we can only give the general description of such process, which is nevertheless also not the focus of the current study.

Moreover, the stretching of hydrogels can also influence the pore structure as discussed in reported work (*Soft Materials*, **2022**, 20(1): 99-108). Usually, the pore structure in hydrogels without pre-stretching is loose and the size distribution of the pore structure is uneven. After pre-stretching, the pore size of the hydrogel became smaller, the pore walls became thinner, and the pore structure became denser.

Supplementary Fig. 36. BJH Pore Size Distribution (PSD) of samples and pore size peak positions for various xerogel samples. The total pore volume was calculated at a P/P_0 of 0.93.

Supplementary Fig. 37. Densities of different parts in the xerogel fibers. The buffer samples were taken as examples, assuming that the samples were perfect cylinders with only two different parts inside. The outer layer was removed mechanically. By recording the mass and diameter of the xerogel samples, the densities of different parts were calculated.

10. Authors should estimate the three evaporation speeds, low, medium, and high is too ambiguous.

Response: Thanks for the valuable comments. To address this point, we calculated the evaporation speed of different media in bulk solutions. Typically, the free liquid drops were placed on the glass slide. The evaporation area is kept constant (constant contact radius mode). The degree of air drying completion was defined as evaporated mass/original mass of the drops. When the value reaches 1, it means that the evaporation process is complete.

Supplementary Fig. 1. The evaporation speeds of different media in air drying process.

From the above experimental data, it is clear that the difference in evaporation rate between these three media follows the theoretical expectation.

What's more, by numerical analysis we obtained the change of water content inside the porous domain with the change of evaporation time. The tendency of these samples in the dynamic process also follows the theoretical prediction. We hope these results can help to understand the difference between various mediums in air drying process.

Supplementary Fig. 15. The roughly quantitative description of different evaporation speeds due to different mediums. The residual water content indicates the ratio of water inside porous domain at certain time point vs original water content. Data was obtained by simulation.

11. In general I found that the montage of schematics and images within Figures 1,2,3, and 4 difficult to “see”. They schematics and images are too small to see the meaningful information that the authors refer to in the text. Thus in makes these figures in effective in sharing information. **Response:** We thank the reviewer for attentive comments. We tried our best to improve the quality of the figures as shown above and in the revised manuscript, and hope they are more clear now.

12. In the conclusion the authors state “ Our work verified the inner connection between soft matters and living things as well”. This work did not show any results that link these two aspects. As I mentioned in issue #1.... the authors should remove all

Response: Thanks for the suggestive comments. We have carefully revised the description in the manuscript and try our best to avoid any misleading information or deceptive relationship between biosynthesis process/ this study.

Other aspect the authors should consider addressing.

13. line 322-323. should be more specific as to "relatively" solid vs liquid. If the hydrogel was a liquid, then how can measure mechanical properties, how could it be stretched.

Response: Thanks for the suggestive comments. We are sorry for any misleading statement here. In classical physics the distinction between a solid and a liquid was based on rheological properties: the solid was elastic and obeyed Hooke’s law, whereas a liquid was viscous and obeyed Newton’s law of constant viscosity (*Contemp. Phys.*, **1968**, 9(6): 537-548). This could be generally used to differentiate solid and liquid-like materials.

Surely, this description will become confusing when solid-like and liquid-like materials were systematically investigated, in particular towards those materials that possessed both elastic and viscous properties. An example is the classic toy Silly Putty. The viscoelasticity of Silly Putty is evidenced by the way it bounces off a table like a rubber ball when interacting quickly with the surface and flows like a thick syrup when left alone for a long time (*Phys. Today*, **2018**, 71 (7), 34–40). Surely, the hydrogel is a viscoelastic solid (or elastic liquid) as well, so-called because it simultaneously exhibits both solid-like (elastic) and liquid-like (viscous) behaviors (Supplementary Fig. 5). The hydrogel can store the elastic energy when external force was applied. This is the reason why relatively “liquid” hydrogel can be applied for uniaxial tensile test or other mechanical properties. Thus, in essence, whether the material is liquid or solid is highly dependent on the time scale over which it is observed.

Connected to the current work, the Deborah number (De) is used to defined solid or liquid for practical purpose (*Phys. today*, **1964**, 17(1): 62). The observation time was covered by De and the difference between solids and fluids can be more precisely defined by the magnitude of De . when the observation time is very large, or, conversely, if the time of relaxation of the material under observation is very small, we see the material flowing ($De < 1$). On the other hand, if the time of relaxation of the material is larger than observation time, the material can be regarded as a solid ($De > 1$). Based on these considerations, the practical behavior of hydrogel was described as relatively liquid or solid in the manuscript.

14. Lines 348 to 353. It is unclear how this hydrogel is different from the one described in lines 339 to 347.

Response: We thank the reviewer for attentive comments. As for lines 348 to 353, the composition of hydrogel was totally the same as that described in lines 339 to 347. The main difference is the treatment process. For the samples described in lines 339 to 347, The hydrogel was stretched and then left to air dry for a period of time (Range of 0-10 min). The transition from homogeneity to heterogeneity was then demonstrated at different observation times in the air drying process. In contrast, the as-prepared virginal hydrogel (without stretch and air drying) was observed, and only the typical pore structure was found. All these experimental data showed that it was the transition from homogeneity to heterogeneity that occurred during the air drying process, rather than intrinsic heterogeneity within the hydrogel.

15. line 499.... unclear what deformation speed is and this does not seem to match the units of (1/s).

Response: We thank the reviewer for the suggestive comments. The deformation speed was defined as $dd\lambda$, as described in Supplementary Fig. 5. Since the variable λ is dimensionless, the unit of deformation speed should be s^{-1} . The description was widely used to describe the change of materials (*Biomaterials*, **2001**, 22(8): 799-806; *Mater. Sci. Eng. A*, **2003**, 350(1-2): 63-69; *Proc. Natl. Acad. Sci.*, **2021**, 118(14): e2014694118). The need to introduce such a deformation rate is to avoid the influence of sample dimension. To clearly indicate the factor, we use the words "initial strain rate" to replace "deformation rate".

One Aspect I liked: The supporting document had a lot of information, figures and table... this was wonderful and helped me greatly in understanding this work.

Response: We are very grateful for the reviewer's recognition of the research. We appreciate the reviewers for the constructive comments and efforts that have enabled us to greatly improve our manuscript. We believe that the novelty and impact are now more clear. Based on the reviewers' comments, we have made extensive alterations to the structure, format, presentation, and analysis of our findings. The supporting document was also promoted with more detailed information.

REVIEWERS' COMMENTS

Reviewer #1 (Remarks to the Author):

The authors have substantially improved the support for the central findings in this work and the manuscript is now suitable for publication in Nature Communications. I would still caution that the authors, in collaboration with the editors, should ensure that the conclusions do not overstep the bounds of what the interesting data here shows, as the generalization of these phenomena remains an open question.

Reviewer #2 (Remarks to the Author):

The concept of using Deborah number-dependent transition to better understand gelation, the resulting structure and resulting properties of CNC-polymer hydrogels was novel, and the results are noteworthy.

The authors have addressed all my comments/issues I had from my review of the original submitted manuscript. As a result, I no longer see any flaws in the data analysis, interpretation and conclusions. The methodology is sound, meeting the standards in the field, there is sufficient details given in the methodology to reproduce the work. The work supports the conclusions and claims made.

Reviewer #1 (Remarks to the Author):

The authors have substantially improved the support for the central findings in this work and the manuscript is now suitable for publication in Nature Communications. I would still caution that the authors, in collaboration with the editors, should ensure that the conclusions do not overstep the bounds of what the interesting data here shows, as the generalization of these phenomena remains an open question.

Response: We thank the reviewer for reviewing this study with very positive comments. We are very grateful for recommending immediate publication of this study in Nature Communications.

Regarding the concern and advice by the **Reviewer #1**, we carefully check the text in the manuscript and make the appropriate revision, please see the highlighted text in the manuscript, so as to present the clear and accurate description and at the same time not to overinterpret the research work.

Reviewer #2 (Remarks to the Author):

The concept of using Deborah number-dependent transition to better understand gelation, the resulting structure and resulting properties of CNC-polymer hydrogels was novel, and the results are noteworthy.

The authors have addressed all my comments/issues I had from my review of the original submitted manuscript. As a result, I no longer see any flaws in the data analysis, interpretation and conclusions. The methodology is sound, meeting the standards in the field, there is sufficient details given in the methodology to reproduce the work. The work supports the conclusions and claims made.

Response: We appreciate **Reviewer #2** for reviewing this study with very positive and attentive comments. We are pleased to learn that the reviewer well acknowledged the importance and novelty of this study. We thank the **Reviewer #2** for recommending publication of this study in Nature Communications.